# Event-Customized Image Generation

**Zhen Wang** [1]   **Yilei Jiang** [1]   **Dong Zheng** [1]   **Jun Xiao** [1]   **Long Chen** [2]

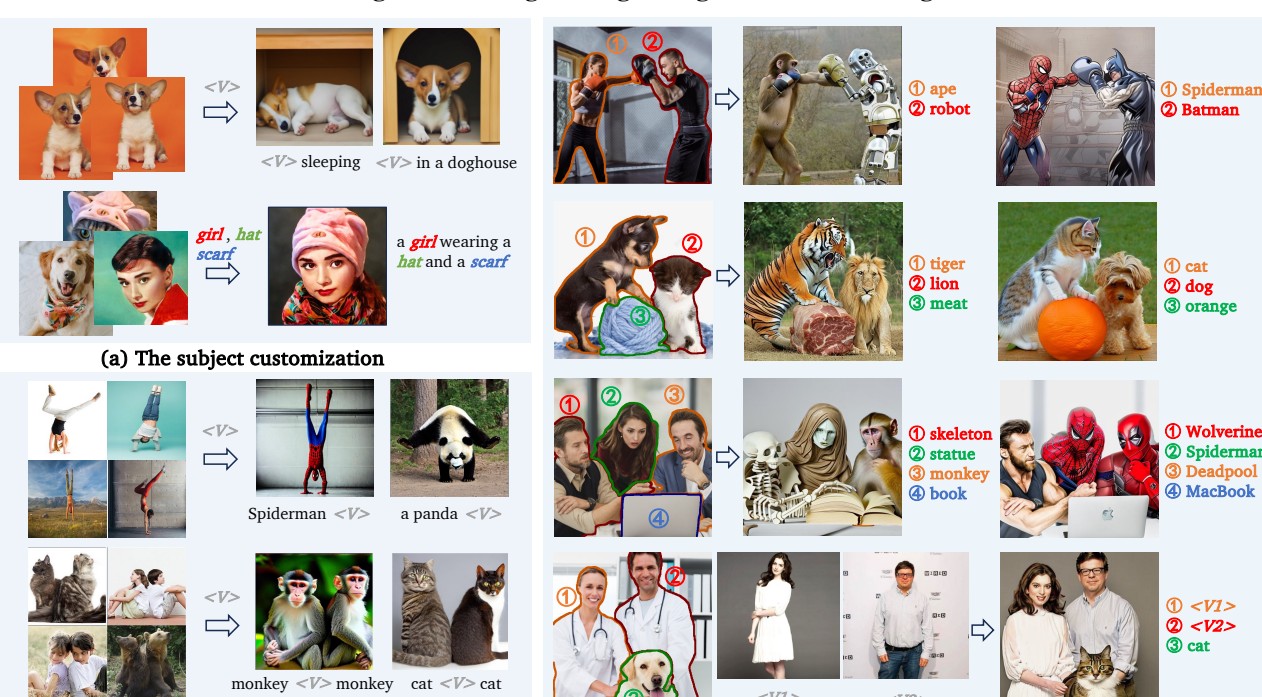

Figure 1: **Customized Image Generation.** (a) Generating customized images with given *subjects* in new contexts. (b) Generating customized images with co-existing basic *action* or *interaction* in given images. (c) Generating customized images for complex *events* with various target entities. Different colors and numbers show associations between reference entities and corresponding target prompts.

## Abstract

Customized image generation has raised significant attention due to its creativity and novelty. With impressive progress achieved in *subject* customization, some pioneer works further explored the customization of *action* and *interaction* beyond entity (*e.g.*, human and object) appearance. However, these approaches only focus on basic actions and interactions between two entities, and their effects are limited by insufficient "exactly same" reference images. To extend the customiza-

tion to more complex scenes, we propose a new task: ***event-customized image generation***. Given a single reference image, we define the "event" as all specific actions, poses, relations, or interactions between different entities in the scene. This task aims at accurately capturing the complex event and generating customized images with various target entities. To solve this task, we proposed a novel training-free method: **FreeEvent**. Specifically, FreeEvent introduces two extra paths alongside the general diffusion denoising process: 1) Entity switching path: it applies cross-attention guidance and regulation for target entity generation. 2) Event transferring path: it injects the spatial feature and self-attention maps from the reference image to the target image for event generation. We further collected two new evaluation benchmarks. Extensive experiments have demonstrated the effectiveness of FreeEvent.

---

[1]Zhejiang University, Hangzhou, China [2]The Hong Kong University of Science and Technology, Hong Kong, China. Work was done when Zhen Wang visited HKUST. Correspondence to: Long Chen <longchen@ust.hk>.

*Proceedings of the 42nd International Conference on Machine Learning*, Vancouver, Canada. PMLR 267, 2025. Copyright 2025 by the author(s).

# 1. Introduction

Recently, large-scale pre-trained diffusion models (Dhariwal & Nichol, 2021; Nichol et al., 2022; Ramesh et al., 2022; Rombach et al., 2022; Saharia et al., 2022) have demonstrated remarkable success in generating diverse and photorealistic images from text prompts. Leveraging these unparalleled creative capabilities, a novel application — customized image generation (Gal et al., 2023a; Ruiz et al., 2023; Chen et al., 2024a) — has gained increasing attention for generating user-specified concepts. Significant progress has already been made in subject-customized image generation (Ye et al., 2023; Chen et al., 2024c). As shown in Figure 1(a), given a set of user-provided subject images, existing methods can accurately capture the unique appearance features of each subject (*e.g.*, `corgi`) with a special identifier token, enabling creative rendering in new and diverse scenarios. Moreover, they can seamlessly integrate multiple subjects into cohesive compositions, preserving distinctive characteristics while adapting them to novel contexts.

Beyond the appearance of different entities (*i.e.*, humans, animals, and objects) in the images, pioneering approaches have been developed to customize the user-specified actions (Huang et al., 2024a), interactive relations (Huang et al., 2024b) and poses (Jia et al., 2024) between the entities. As shown in Figure 1(b), these methods attempt to capture the single-entity action (*e.g.*, `handstand`) or interactions (*e.g.*, `back to back`) between two entities that co-exist in the given reference images and transfer them to the synthesis of action- or interaction-specific images with new entities.

However, for real-world scenes that typically involve multiple entities with more complex interactions (*e.g.*, Figure 1(c), row 3: `three humans are discussing in front of a computer with different poses`), these works (Huang et al., 2024b;a; Jia et al., 2024) still face notable limitations. **1) Simplified Customization.** Current action customization (Huang et al., 2024a) focuses solely on the basic actions of a single person. Similarly, interaction customizations (Huang et al., 2024b; Jia et al., 2024) are limited to basic interactive relations or poses between just two entities. There is a lack of exploration into more complex and diverse actions or interactions that involve multiple humans, animals, and objects. Additionally, while these methods typically perform well when generating images with the same type of entity (*e.g.*, all monkeys or all cats), they struggle when faced with more diverse and complex entities and their combinations. These narrow focuses and limitation on entity generation have strictly limited their abilities to customize more complex and diverse scenes with creative content. **2) Insufficient Data.** To capture specific actions or interactions, existing methods (Huang et al., 2024b;a; Jia et al., 2024) tend to represent them by learning corresponding identifier tokens, which can be further used for generating new images. However, for each action, or interaction, these training-based processes typically require a set of reference images (*e.g.*, 10 images) paired with corresponding textual descriptions across different entities. Unfortunately, each action or interaction is highly unique and distinctive, *i.e.*, gathering images that depict the exact same action or interaction is challenging. As shown in Figure 1(b), there are still significant differences in the same action (*e.g.*, `handstand`) between different reference images, which thus compromises the accuracy of learned tokens, leading to inconsistencies in action between generated images. This insufficient data issue for identical action or interaction has severely limited the practicality and generalizability of these methods.

To address these limitations and extend customized image generation to more complex scenes, we propose a new and meaningful task: **event-customized image generation**. Given a single reference image, we define the "event" as all actions and poses of each single entity, and their relations and interactions between different entities. As shown in Figure 1(c), event customization aims to accurately capture the complex and diverse event from the reference image to generate target images with various combinations of target entities. Since it only needs one single reference image, the event customization also eliminates the need for collecting "exactly same" reference images.

To solve this challenging task, we proposed a novel *training-free* event customization method, denoted as **FreeEvent**. Based on the two main components of the reference image, *i.e.*, entity and event, FreeEvent decomposes the event customization into two parts: 1) Switching the entities in the reference image to target entities. 2) Transferring the event from the reference image to the target image. Following this idea, alongside the general denoising process of diffusion generation, we designed two extra paths: entity switching path and event transferring path. Specifically, entity switching path guides the localized layout of each target entity for entity generation. Event transferring path further extracts the event information from the reference image and then injects it into the denoising process to generate the specific event. Through this direct guidance and injection, FreeEvent offers a significant advantage over existing methods by eliminating the need for time-consuming training. Furthermore, as shown in Figure 1(c), FreeEvent can also serve as a plug-and-play framework to combine with subject customization methods, generating creative images with both user-specified events and subjects.

Moreover, as a pioneering effort in this direction, we also collected two evaluation benchmarks from the existing dataset (*i.e.*, SWiG (Pratt et al., 2020) and HICO-DET (Chao

et al., 2015)) and the internet for event-customized image generation, dubbed **SWiG-Event** and **Real-Event**, respectively. Both benchmarks include reference images featuring diverse events and entities, along with manually crafted target prompts. Extensive experiments demonstrate that our approach achieves state-of-the-art performance, enabling more complex and creative customization with enhanced practicality and generalizability.

In summary, we make several contributions in this paper: 1) We propose the novel event-customized image generation task, which extends customized image generation to more complex scenes in real-world applications. 2) We propose FreeEvent, the first training-free method for event customization, which can be further combined with subject customization methods for more creative and generalizable customizations. 3) We collect two evaluation benchmarks for event-customized image generation, and FreeEvent achieves outstanding performance compared with existing methods.

## 2. Related Work

**Text-to-Image Diffusion Generation.** Diffusion models (Ho et al., 2020; Nichol & Dhariwal, 2021; Song et al., 2021) have emerged as a leading approach for image synthesis. The text-to-image diffusion models (Nichol et al., 2022; Ramesh et al., 2022; Saharia et al., 2022) further inject user-provided text descriptions into the diffusion process via pre-trained text encoders. After trained on large-scale text-image pairs, they have shown great success in text-to-image generation. Different from these models that operate the diffusion process on pixel space, the latent diffusion models (LDMs) (Rombach et al., 2022) propose to perform it on latent space with enhanced computational efficiency. Besides, existing works (Hertz et al., 2023; Tumanyan et al., 2023; Cao et al., 2023; Alaluf et al., 2024) have discovered the spatial feature and attention maps in LDMs contain localized semantic information of the image and the layout correspondence between textual conditions. As a result, these features and attention maps have been utilized to control the layout, structure, and appearance in text-to-image generation. This can be achieved either through a plug-and-play feature injection (Tumanyan et al., 2023; Xu et al., 2024; Lin et al., 2024) or by computing specific diffusion guidance (Epstein et al., 2023; Mo et al., 2024) for generation. In this paper, we utilize the pre-trained LDM Stable Diffusion (Rombach et al., 2022) as our base model.

**Subject Customization.** This task aims to generate customized images of user-specified subjects. Current mainstream subject customization works mainly focus on 1) Single subject customization, including learning specific identifier tokens (Gal et al., 2023a), finetuning the text-to-image diffusion model (Ruiz et al., 2023; 2024), introducing layer-wise learnable embeddings (Voynov et al., 2023)

and training large-scale multimodal encoders (Gal et al., 2023b; Li et al., 2024). 2) Multi-subject composition, including cross-attention modification (Tewel et al., 2023), constrained model fine-tuning (Kumari et al., 2023), layout guidance (Liu et al., 2023), and gradient fusion of each subject (Gu et al., 2024). In conclusion, these works are tailored to capture the appearance of the entities in the image, without considering the customization of actions or poses.

**Action and Interaction Customization.** They aim to generate customized images with co-existing actions or interactions in user-provided reference images. ReVersion (Huang et al., 2024b) first proposes to customize specific interactive relations by optimizing the learnable relation tokens. ADI (Huang et al., 2024a) makes progress in customizing specific actions for a single subject. And a following work (Jia et al., 2024) further extends it to learning interactive poses between two individuals. However, all these works only focus on simplified customization of some basic actions and interactions, and their effect is strictly limited by the insufficient data of reference images. In contrast, our proposed event customization only requires one reference image, and our training-free framework can achieve effective customization of complex events with various creative target entities.

## 3. Methods

### 3.1. Preliminary

**Latent Diffusion Model (LDM).** Generally, LDMs include a pretrained autoencoder and a denoising network. Given an image $x$, the encoder $\mathcal{E}$ maps the image into the latent code $z_0 = \mathcal{E}(x)$, where the forward process is applied to sample Guassian noise $\epsilon \sim \mathcal{N}(0, \mathbf{I})$ to it to obtain $z_t = \sqrt{\bar{\alpha}_t} z_0 + \sqrt{1 - \bar{\alpha}_t} \epsilon$ from time step $t \sim [1, T]$ with a predefined noise schedule $\bar{\alpha}$. While the backward process iteratively removes the added noise on $z_t$ to obtain $z_0$, and decodes it back to image with the decoder $x = \mathcal{D}(z_0)$. Specifically, the diffusion model is trained by predicting the added noise $\epsilon$ conditioned on time step $t$ and possible conditions like text prompt P. The training objective is formulated as:

$$\mathcal{L}_{\text{LDM}} = \mathbb{E}_{z \sim \mathcal{E}(x), P, \epsilon \sim \mathcal{N}(0,1), t} \left[ \|\epsilon - \epsilon_\theta(z_t; t, P)\|_2^2 \right]. \quad (1)$$

where $\epsilon_\theta$ is the denoising network.

**Diffusion Guidance.** The diffusion guidance modifies the sampling process (Ho et al., 2020) with additional score functions to guide it with more specific controls like object layout (Xie et al., 2023; Mo et al., 2024) and attributes (Epstein et al., 2023; Bansal et al., 2023). We express it as

$$\hat{\epsilon}_t = \epsilon_\theta(z_t; t, P) - s \cdot \mathbf{g}(z_t; t, P), \quad (2)$$

where $\mathbf{g}$ is the energy function and $s$ is a parameter that controls the guidance strength.

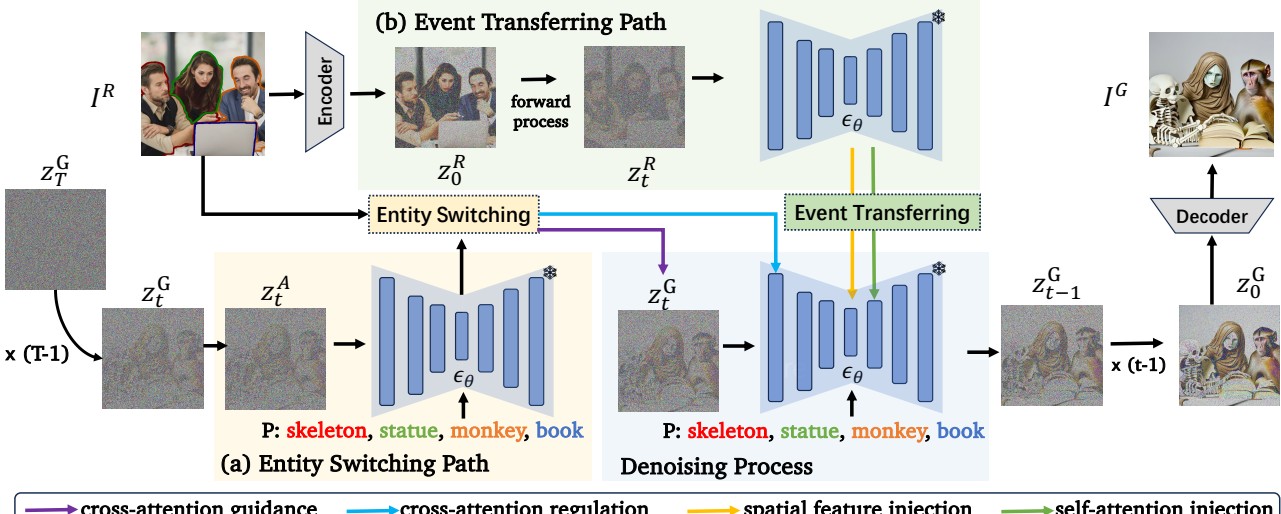

Figure 2: **The overview of pipeline.** Given the reference image, the event customization is overall a general diffusion denoising process with two extra paths. 1) The entity switching path guides the generation of each target entity through cross-attention guidance and regulation 2) The event transferring path injects the spatial features and self-attention maps from the reference image to the denoising process. The final $z_0^G$ is then transformed back to target image $I^G$ by the decoder.

## 3.2. Task Definition: Event-customized Generation

In this section, we first formally define the event-customized image generation task. Given a reference image $I^R$ involves $N$ reference entities $\mathrm{E}^R = \{R_1, \ldots, R_N\}$, we define the "event" as the specific actions and poses of each single reference entity, and the relations and interactions between different reference entities. Together we have the entity masks $\mathrm{M} = \{m_1, \ldots, m_N\}$, where $m_i$ is the mask of its corresponding entity $R_i$. The event-customized image generation task aims to capture the reference event, and further generate a target image $I^G$ under the same event but with diverse and novel target entities $\mathrm{E}^G = \{G_1, \ldots, G_N\}$ in the target prompt $\mathrm{P} = \{w_0, \ldots, w_N\}$, where $w_i$ is the description of the target entity $G_i$, and each target entity $G_i$ should keep the same action or pose with its corresponding reference entity $R_i$. As the example shown in Figure 2, given the reference image with four reference entities (*e.g.*, three people and one object), the event-customization aims to capture the complex reference event and generate the target image with a novel combination of different target entities (*e.g.*, skeleton, statue, monkey, book).

## 3.3. Approach

**Overview.** We now introduce the proposed training-free event customization framework FreeEvent. Specifically, we decompose the event-customized image generation into two parts, 1) generating target entities (*i.e.*, switching each reference entity to target entity), and 2) generating the same reference event (*i.e.*, transferring the event from the reference image to the target image). Following this idea, we design two extra paths for the diffusion denoising process

of event customization, denoted as the entity switching path and the event transferring path, respectively. Generally, as shown in Figure 2, the generation of $I^G$ starts by randomly initializing the latent $z_T^G \sim \mathcal{N}(0, \mathbf{I})$, and iteratively denoise it to $z_0^G$. During this denoising process, the entity switching path guides the generation of each target entity through cross-attention guidance and regulation based on the target prompt P and reference entity masks M. The event transferring path extracts the spatial features and self-attention maps from the reference image $I^R$, and then injects them to the denoising process. The final $z_0^G$ is then transformed back to the target image $I^G$ by the decoder.

**U-Net Architecture** The Stable Diffusion (Rombach et al., 2022) utilizes the U-Net architecture (Ronneberger et al., 2015) for $\epsilon_\theta$, which contains an encoder and a decoder, where each consists of several basic encoder/decoder blocks, and each encoder/decoder block further contains several encoder/decoder layers. Specifically, as shown in Figure 3(a), each U-Net encoder/decoder layer consists of a residual module, a self-attention module, and a cross-attention module. For block $b$, layer $l$, and timestep $t$, the residual module produces the spatial feature of the image as $\mathbf{f}$. The self-attention module produces the self-attention map as $\mathbf{SA} = \mathrm{Softmax}(\frac{\mathbf{Q}_s \mathbf{K}_s^{\mathrm{T}}}{\sqrt{d}})$, where $\mathbf{Q}_s$ and $\mathbf{K}_s$ are query and key features projected from the visual features. For text-to-image generation, the cross-attention module further produces the cross-attention map between the text prompt P and the image as $\mathbf{CA} = \mathrm{Softmax}(\frac{\mathbf{Q}_c \mathbf{K}_c^{\mathrm{T}}}{\sqrt{d}})$, where $\mathbf{Q}_c$ is the query features projected from the visual features, and $\mathbf{K}_c$ is the key features projected from the textual embedding of P.

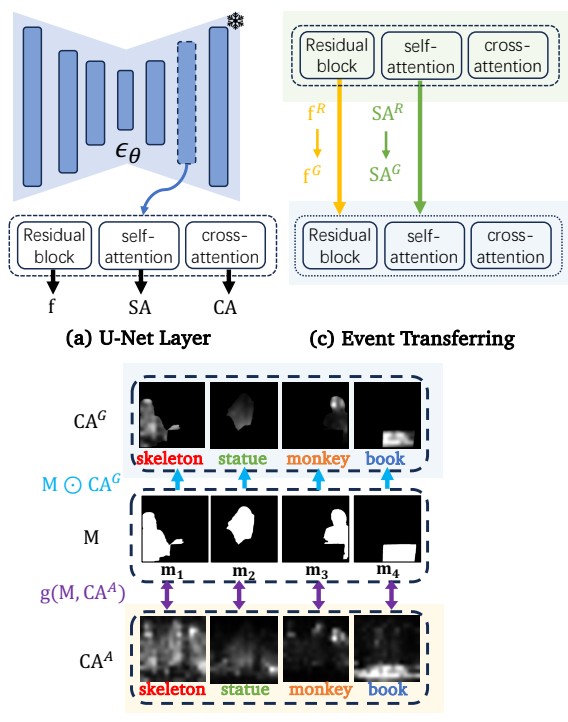

(a) U-Net Layer     (c) Event Transferring

(b) Entity Switching

Figure 3: **(a)** The architecture of the U-Net layer. **(b)** The process of cross-attention guidance and regulation. **(c)** The process of spatial feature and self-attention injection.

**Entity Switching Path.** This path aims on generating target entities $E^G = \{G_1, \ldots, G_N\}$ in $I^G$ by switching each reference entity $R_i$ to $G_i$ based on the target prompt P and reference entity masks M. And the key is to ensure each target entity $G_i$ is generated at the same location as their corresponding reference entity $R_i$ and avoid the appearance leakage between different entities. Inspired by prior works (Hertz et al., 2023; Chen et al., 2024b) that utilize the cross-attention maps to control the layout of text-to-image generation, we apply the cross-attention guidance and regulation to achieve the entity switching.

As shown in Figure 2(a), at the timestep $t$ of the denoising process, we first obtain the latent for entity switching as $z_t^A = z_t^G$, we then input $z_t^A$ together with the target prompt P into the U-Net, and calculate the cross-attention maps as $\mathbf{CA}^A$. Then, we introduce an energy function to bias the cross-attention of each token $w_i$ as (*cf.*, Figure 3(b)):

$$\mathbf{g}(\mathrm{CA}_i^A, m_i) = (1 - \frac{\mathrm{CA}_i^A * m_i}{\mathrm{CA}_i^A})^2 \qquad (3)$$

where $\mathrm{CA}_i^A$ is the cross-attention map of token $w_i$. Optimizing this function encourages the cross-attention maps of each target entity $G_i$ to obtain higher values inside the corresponding area specified by $m_i$, which further guides the localized layout of each target entity. We calculate the gradient of this guidance via backpropagation to update

latent $z_t^G$:

$$z_t^G = z_t^A - \sigma_t^2 \eta \bigtriangledown_{z_t^A} \sum_{i \in N} \mathbf{g}(\mathrm{CA}_i^A, m_i) \qquad (4)$$

where $\eta$ is the guidance scale and $\sigma_t = \sqrt{(1 - \bar{\alpha}_t / \bar{\alpha}_t)}$. Additionally, to avoid the appearance leakage between each target entity, we further regulate the cross-attention map of each token within its corresponding area. Specifically, for cross-attention maps $\mathbf{CA}^G$ calculated at timestep $t$ during the denoising process, we have:

$$\mathrm{CA}_i^G = m_i \odot \mathrm{CA}_i^G \qquad (5)$$

where $\mathrm{CA}_i^G$ is the cross-attention map of token $w_i$.

**Event Transferring Path.** This path aims to extract the specific reference event from the reference image $I^R$, including the action, pose, relation, or interactions between each reference entity, and transferring them to the target image $I^G$. Meanwhile, from the perspective of image spatial information, the event is essentially the structural, semantic layout, and shape details of the image. Thus, based on the observation that the spatial features and self-attention maps can be utilized to control the image layout and structure (Tumanyan et al., 2023; Xu et al., 2024; Lin et al., 2024), we perform spatial feature and self-attention map injection to achieve the event transferring.

Specifically, as shown in Figure 2(b) we first get the latent code of the reference image $z_0^R = \mathcal{E}(I^R)$, and at each time step $t$ during the denoising process, we obtain $z_t^R$ via the diffusion forward process. We then input $z_t^R$ into the U-Net to extract the spatial features and self-attention maps of the reference image as $\mathbf{f}^R$ and $\mathbf{SA}^R$. Parallelly, for the denoising process, we input $z_t^G$ together with the target prompt P into the U-Net, and calculate the spatial features and self-attention maps for the generated image as: $\mathbf{f}^G$ and $\mathbf{SA}^G$. Then, as shown in Figure 3(c), we perform the injection by directly replacing corresponding spatial features and self-attention maps:

$$\mathbf{f}^G \leftarrow \mathbf{f}^R \quad \text{and} \quad \mathbf{SA}^G \leftarrow \mathbf{SA}^R. \qquad (6)$$

***Highlights.*** By applying cross-attention guidance and regulation on each text token, our attention-guided entity switching can also be used to generate target entities of user-specified subjects, *i.e.*, represented by specific identifier tokens. Thus, our framework can be easily combined with subject customization methods to generate creative images with both customized events and subjects.

## 4. Experiments

### 4.1. Experimental Setup

**Evaluation Benchmarks.** In order to provide sufficient and suitable conditions for both quantitative and qualitative

| Model | Image Retrieval | | | Verb Detection | | | Image Similarity | | | CLIP-T↑ | FID↓ |
|---|---|---|---|---|---|---|---|---|---|---|---|
| | R@1↑ | R@5↑ | R@10↑ | T-1↑ | T-5↑ | T-10↑ | CLIP-I↑ | DreamSim↑ | DINO↑ | | |
| ControlNet | 10.64 | 26.12 | 36.82 | 10.66 | 23.98 | 31.28 | 0.6009 | 0.3714 | 0.2599 | 0.2198 | 70.45 |
| MIGC | 10.90 | 27.00 | 37.64 | 10.62 | 26.14 | 35.04 | 0.6456 | 0.3772 | 0.2467 | 0.2145 | 49.81 |
| BoxDiff | 8.60 | 22.48 | 32.08 | 5.58 | 14.52 | 19.42 | 0.5838 | 0.3135 | 0.2099 | 0.2153 | 68.49 |
| **FreeEvent** | **41.12** | **63.02** | **72.74** | **34.10** | **62.04** | **71.82** | **0.7044** | **0.6282** | **0.5116** | **0.2238** | **29.05** |

Table 1: Performance of our model and state-of-art conditional text-to-image generation models on SWiG-Event. For image retrieval, the R@k represents that among the top-k images with the highest similarity to the target image, its corresponding reference image is included. For verb detection, the T-K represents the top-k detection accuracy.

comparisons on this new task, we collect two new benchmarks[1]. 1) For quantitative evaluation, we present *SWiG-Event*, a benchmark derived from SWiG (Pratt et al., 2020) dataset, which comprises 5,000 samples with various events and entities, *i.e.*, 50 kinds of different actions, poses, and interactions, where each kind of event has 100 reference images, and each reference image contains 1 to 4 entities with labeled bounding boxes and nouns. 2) For qualitative evaluation, we present *Real-Event*, which comprises 30 high-quality reference images from HICO-DET (Chao et al., 2015) and the internet with a wide range of events and entities (*e.g.*, animal, human, object, and their combinations). We further employ Grounded-SAM (Kirillov et al., 2023; Ren et al., 2024) to extract the mask of each entity.

**Baselines.** We compared several kinds of SOTA baselines. For conditioned text-to-image generation baselines, we compared with training-based method ControlNet (Zhang et al., 2023), MIGC (Zhou et al., 2024), and training-free method BoxDiff (Xie et al., 2023). For localized editing baselines, we compared with training-free methods PnP (Tumanyan et al., 2023) and MAG-Edit (Mao et al., 2024). For customization baselines, we compared with training-based methods Dreambooth (Ruiz et al., 2023) and ReVersion (Huang et al., 2024b).

**Implementation Details.** We use Stable Diffusion v2-1-base as base model for all methods. Images are generated with a resolution of 512x512 on a NVIDIA A100 GPU[1].

### 4.2. Quantitative Comparisons

We compare FreeEvent with state-of-the-art conditional text-to-image generation baselines ControlNet (Zhang et al., 2023), MIGC (Zhou et al., 2024), and BoxDiff (Xie et al., 2023) on the SWiG-Event.

**Setting.** Each reference image in SWiG-Event contains reference entities together with labeled event class, bounding boxes, nouns, and their corresponding masks. Specifically, we construct the target prompt as a list of reference entity nouns, *i.e.*, we ask all the methods to *reproduce* the event of the reference image with the same reference event and same

reference entities. Additionally, ControlNet takes the semantic map merged from the masks as the layout condition. MIGC and BoxDiff take the bounding boxes with labeled entity nouns as the layout condition[1].

**Evaluation.** Our evaluations follow the principle of *whether generated images are aligned or similar with their reference images*, and we apply multiple metrics to evaluate the customization quality of 5,000 target images from different perspectives. 1) Global image similarity: image retrieval performance and similarity scores. We retrieved each target image for its corresponding reference image based on the CLIP score across all the 100 reference images that have the same reference event class. Specifically, we extracted the image feature of each image through a pre-trained CLIP (Radford et al., 2021) visual encoder and calculated the cosine similarities for image retrieval. Besides, we also used the CLIP-I, DreamSim (Fu et al., 2023), and DINO (Oquab et al.) scores to evaluate the image alignment of generated images with their reference images. 2) Event similarity: verb detection performance. We utilized the verb detection model GSRTR (Cho et al., 2021) which was trained on the SWIG dataset to detect the verb class of each generated image, and then calculated the detection accuracy based on the annotations of the reference images (*i.e.*, whether the generated images and their reference images have the same verb class). 3) Entity similarity: CLIP-T (Radford et al., 2021) score. We use the CLIP-T score to evaluate the text alignment of the generated images with text prompts. 4) Standard image generation metric. For a more comprehensive comparison, we used the FID (Heusel et al., 2017) score to evaluate the overall quality of generated images.

**Results.** As shown in Table 1, we can observe: 1) FreeEvent has better retrieval performance than both ControlNet, MIGC, and BoxDiff. This demonstrates that the target images generated by FreeEvent better preserve the overall characteristics of the reference event and entity. 2) FreeEvent also achieves the best verb detection performance, which indicates our method can better preserve the interaction semantics of the generated images. 3) FreeEvent further achieves superior performance over baselines across all similarity scores and standard image generation metrics, indicating our method can generate images with better qualities and

---

[1] Due to limited space, more details/results are in the Appendix.

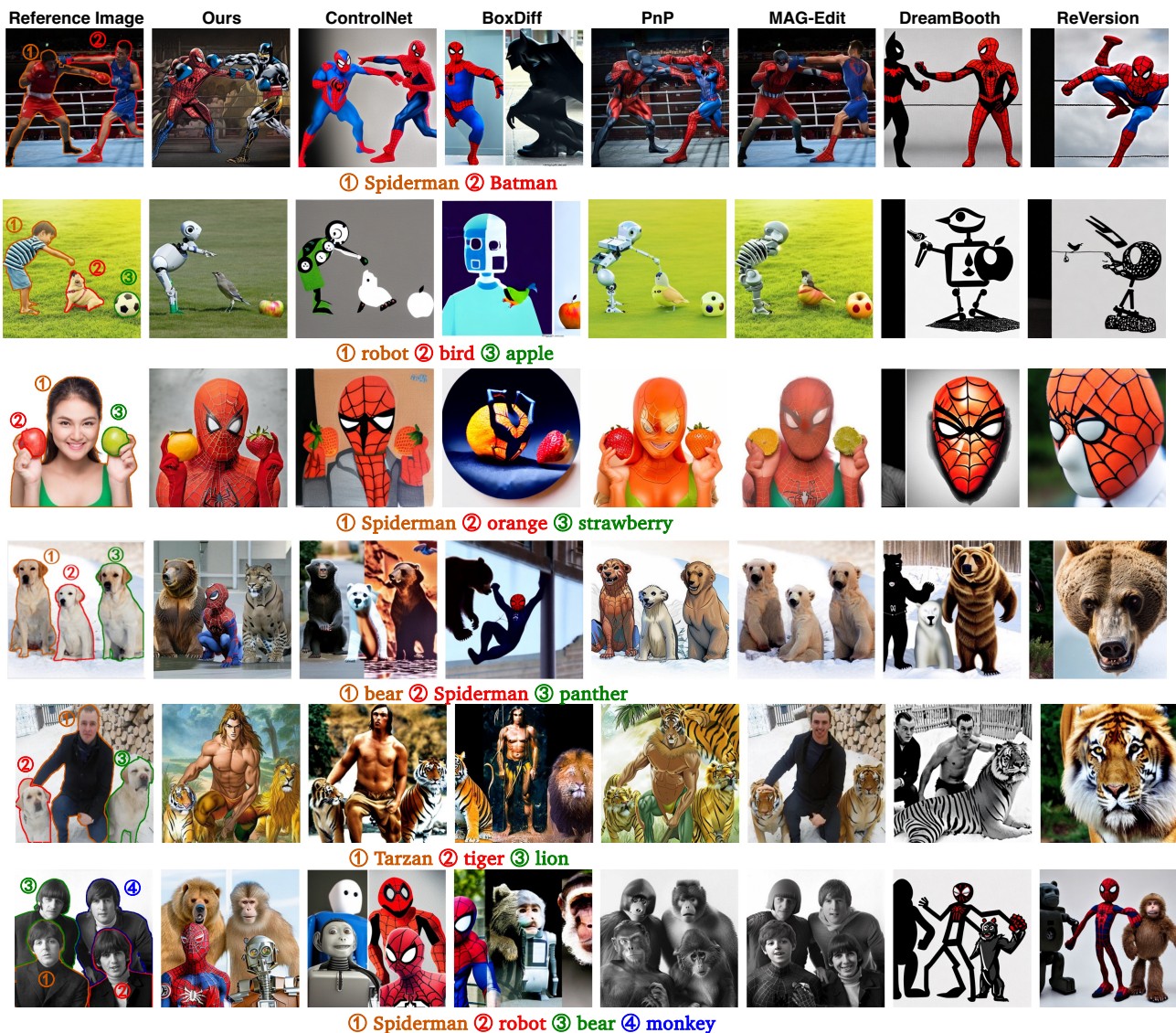

Figure 4: **Comparision of Event Customization.** Different colors and numbers show the associations between reference entities and their corresponding target prompts.

alignment with both the reference images and texts. These results all demonstrate the effectiveness of FreeEvent for event customization.

### 4.3. Qualitative Comparisons

We compare FreeEvent with a wide range of SOTA baselines on the Real-Event[1], including conditioned text-to-image generation method **ControlNet** (Zhang et al., 2023) and **BoxDiff** (Xie et al., 2023), localized image editing method **PnP** (Tumanyan et al., 2023) and **MAG-Edit** (Mao et al., 2024), image customization methods **Dreambooth** (Ruiz et al., 2023) and **ReVersion** (Huang et al., 2024b).

**Setting.** For each reference image in Real-Event, we manually constructed target prompts with various combinations of different target entities. Specifically, ControlNet takes

the semantic map and BoxDiff takes the labeled bounding boxes as the layout conditions. MAG-Edit takes the reference entity masks for localized editing. Dreambooth and ReVersion learn event-specific identifier tokens for text-to-image generation.

**Results.** As shown in Figure 4, we can observe: 1) Conditional text-to-image generation models ControlNet and BoxDiff can only maintain the rough layout of each entity and struggle to capture the detailed action, pose, or interaction between different entities. And they both failed to match the generated entity with the desired target prompt. 2) For localized image editing methods PnP and MAG-Edit, while they can capture the reference event, they both struggle to accurately generate the target entities, and suffer from severe appearance leakage between each target entity

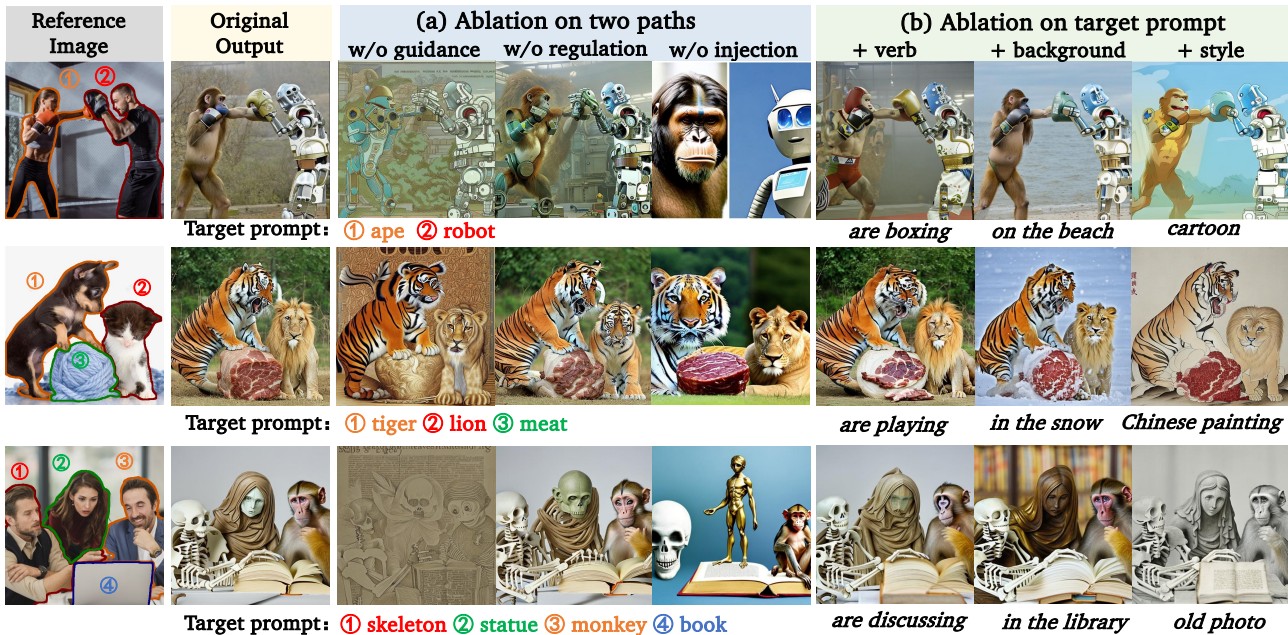

Figure 5: **Ablations of the proposed paths and the target prompt.** The "guidance" and "regulation" denote the cross-attention guidance and cross-attention regulation in entity switching path, respectively. The "injection" denotes the event transferring path.

(*e.g.*, `orange` and `strawberry` in row three, `tiger` and `lion` in row five), and sometimes even failed to edit and output the original content. 3) The subject-customization model Dreambooth and the relation-customization model ReVersion both failed to generate satisfying results. As discussed before, these training-based methods require multiple reference images and are unable to learn the specific event when facing only one reference image. 4) Obviously, our FreeEvent successfully achieves the customization of various complex events with novel combinations of target entities. Meanwhile, the ControlNet and the localized image editing models tend to generate the target entities strictly matching the mask of their corresponding reference entities (*e.g.*, `bird` in row two), which appears very incongruous. On the contrary, the entities generated by FreeEvent not only match the layout of the reference entity but also keep it harmonious. After all, while we use the reference entity mask to guide the generation of each target entity, the cross-attention guidance focuses on directing the overall layout of each target entity and doesn't restrict detailed appearance, allowing for more diverse generation of target entities[1].

### 4.4. Ablations

**Effectiveness of Entity Switching Path and Event Transferring Path.** We first run ablations to verify the effect of two proposed paths during event customization.

*Results.* As the results are shown in Figure 5(a), we can observe: 1) For the entity switching path, removing the cross-attention guidance results in the failure of target entities generation (*e.g.*, the ape, the meat), and removing cross-attention regulation leads to the appearance leakage between entities (*e.g.*, the tiger and lion, the skeleton and statue). 2) After removing the event transferring path, although the target entities can be generated, the reference events are completely lost (*i.e.*, the pose, action, relations, and interactions between each entity). These results all corroborates the effect of two paths in event customization.

**Influence of Different Target Prompts.** Notably, in our paper, the target prompt only contains the nouns of the target entities, we then run the ablations to analyze the influence of different descriptions (*i.e.*, verb, background, style) in the target prompt for event customization.

*Results.* From Figure 5(b) we can observe: 1) Adding *verb* description leads to a certain degree of negative impact on entity appearance (*e.g.*, the head of the ape, the face of the monkey) since these verbs may not be aligned with the model. Besides, accurately describing events in complex scenes can be challenging for users. Therefore, since FreeEvent can already achieve precise extraction and transfer of the reference events, users do not need to describe the specific events in the target prompt, which further demonstrates FreeEvent's practicality. 2) FreeEvent can accurately generate extra contents for the *background* and *style*. Although there may be some detailed changes in the entity's appearance compared to the original output, these do not affect the entity's characteristics or the event. This also

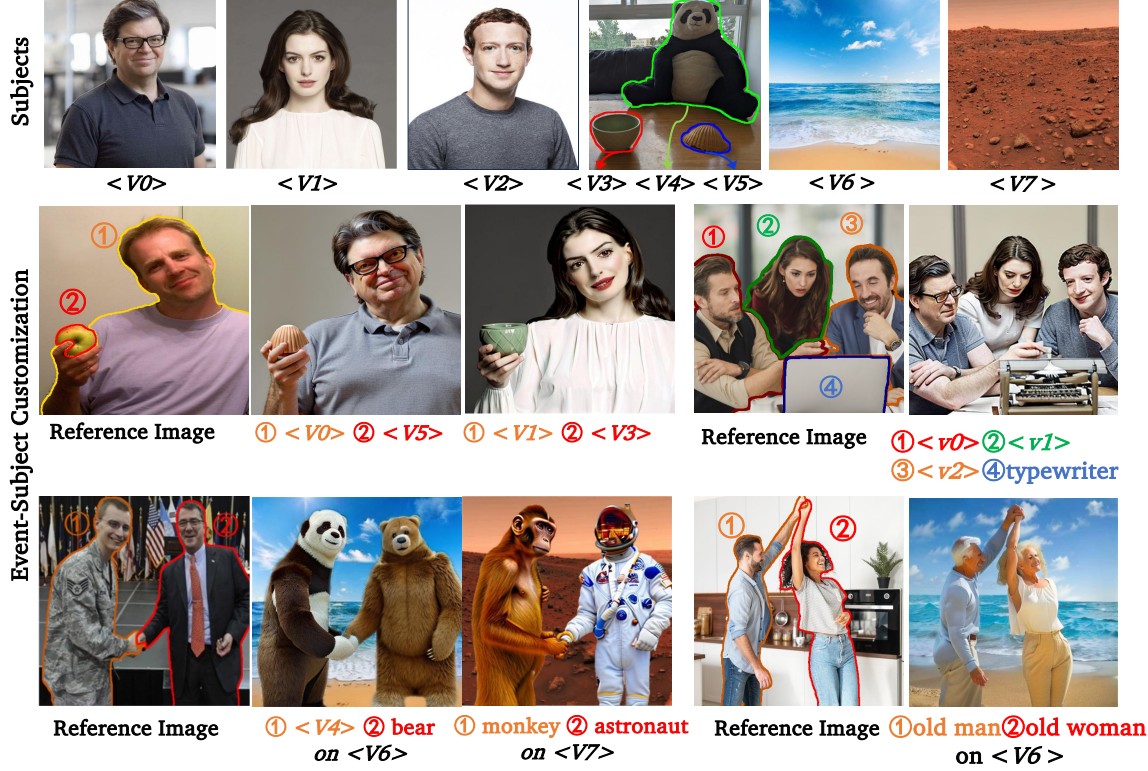

Figure 6: **Results of Event-Subject Customization.** Different colors and numbers show the associations between reference entities and their corresponding target prompts.

| Model | Ours | ControlNet | BoxDiff | PnP | MAGEdit | DreamBooth | ReVersion |
|-------|------|------------|---------|-----|---------|------------|-----------|
| HJ | **50** | 25 | 7 | 26 | 23 | 8 | 3 |

Table 2: Results of the user studies on the Real-Event.

demonstrates FreeEvent's strong generalization capability.

**Combination of Event and Subject Customization.** We further validate the ability of our framework to combine with subject customization methods to generate target entities with user-specified subjects, *i.e.*, represented by identifier tokens. We took the Break-A-Scene model (Avrahami et al., 2023) to learn identifier tokens for subjects and replaced the Stable Diffusion models in Figure 2 with the fine-tuned one.

*Results.* As shown in Figure 6, FreeEvent can effectively generate various given subjects in specific events. Specifically, FreeEvent enables the flexible generation of a wide range of subject concepts (*e.g.*, humans, regular objects, and backgrounds) and their combinations. These results demonstrated the great potential of our framework for Event-Subject customization.

### 4.5. User Study

**Setting.** We conducted user studies on Real-Event to further evaluate the effectiveness of FreeEvent. Specifically, we invited 10 experts and gave them a reference image, a target prompt, and seven target images generated by different models. They are asked to select at least one and up to three images that they believe demonstrate the best results in event customization, taking into account the generation effects of the events and entities, as well as the overall coherence of the images. We prepared 50 trials and asked the experts to give their judgments. The target image which got more than six votes is regarded as human judgment.

**Results.** As shown in Table 2, FreeEvent achieves better performance on human judgments (HJ) compared with all the baseline models.

## 5. Conclusion

In this paper, we proposed a new task: Event-Customized Image Generation. It focuses on the customization of complex events with various target entities. Meanwhile, we proposed the first training-free event-customization framework **FreeEvent**. To facilitate this new task, we also collected two evaluation benchmarks from existing datasets and the internet, dubbed SWiG-Event and Real-Event, respectively. We validate the effectiveness of FreeEvent with extensive comparative and ablative experiments. Moving forward, we are going to 1) extend the event customization into other modalities, *e.g.*, video generation; 2) explore advanced techniques for the finer combination of different customization works, *e.g.*, subject, event, and style customizations.

## Acknowledgements

This work was supported by the National Key Research & Development Project of China (2024YFB3312900), Key R&D Program of Zhejiang (2025C01128), an Fundamental Research Funds for the Central Universities. Long Chen was supported by the Hong Kong SAR RGC Early Career Scheme (26208924), the National Natural Science Foundation of China Young Scholar Fund (62402408), Huawei Gift Fund, and the HKUST Sports Science and Technology Research Grant (SSTRG24EG04).

## Impact Statement

Since FreeEvent can seamlessly integrate with subject customization methods to generate target entities based on user-specified subjects, this capability also raises the same concerns about the potential misuse of pretrained SD models for malicious applications (*e.g.*, Deepfakes) involving real human figures. To address this, it is essential to implement robust safeguards and ethical guidelines, similar to the security measures and NSFW content detection mechanisms already present in existing diffusion models.

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

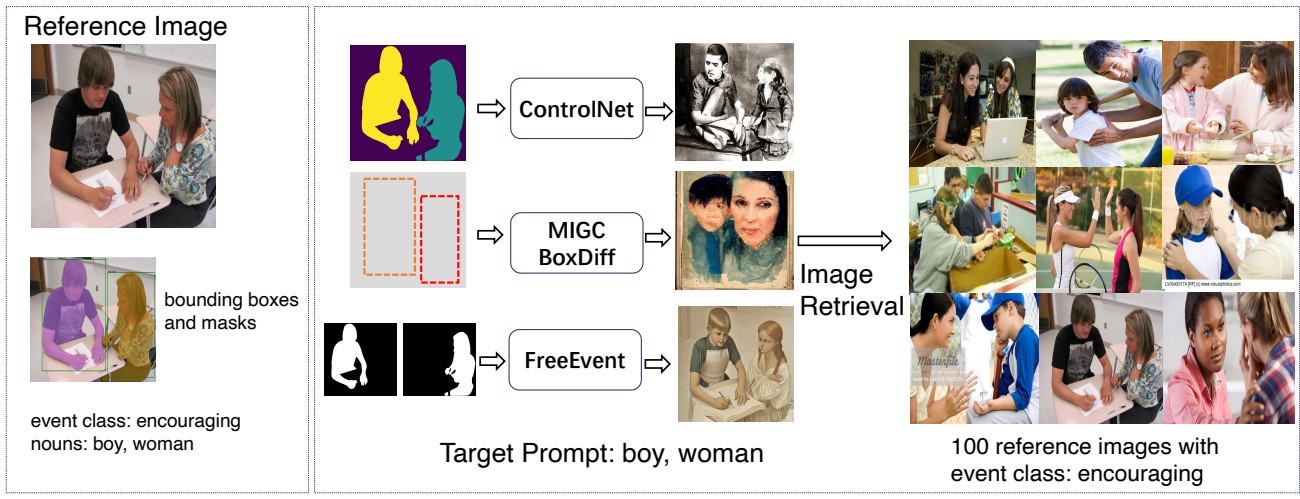

(a) SWiG-Event Sample      (b) Retrieval-based Evaluation

Figure 7: **(a)** The SWiG-Event sample. **(b)** The process of quantitative evaluation and image retrieval.

# Appendix

The Appendix is organized as follows:

- In Sec. A, we show more implementation details.

- In Sec. B, we show more details of the SWiG-Event benchmark and the process of quantitative evaluation and image retrieval.

- In Sec. C, we show quantitative results for path effectiveness.

- In Sec. D, we show the results for attribute generation during event customization.

- In Sec. E, we show more results for event-subject customization comparisions.

- In Sec. F, we provide more discussion about the hyperparameter setting.

- In Sec. G, we provide the discussion of our work's limitations.

- In Sec. H, we show more qualitative comparison results of event customization on the Real-Event.

## A. Implementation Details.

The denoising process was set with 50 steps. For entity switching path, for all blocks and layers containing the cross-attention module, we apply the cross-attention guidance during the first 10 steps. And apply the cross-attention regulation during the whole 50 steps. For event transferring path, we perform spatial feature injection for block and layer at $\{\text{decoder block } 1 : [\text{layer } 1]\}$ during the whole 50 steps. And perform self-attention injection for blocks and layers at $\{\text{decoder block } 1 : [\text{layer } 1, 2], \text{decoder block } 2 : [\text{layer } 0, 1, 2], \text{decoder block } 3 : [\text{layer } 0, 1, 2]\}$ during the first 25 steps. We set the classifier-free guidance scale to 15.0.

## B. Details of SWiG-Event and process of image retrieval.

As shown in Figure 7(a), each SWiG-Event sample consists of a reference image with labeled bounding boxes and masks for each reference entity, the nouns of each reference entity, and the event class. As shown in Figure 7(b), we constructed the target prompt as a list of reference entity nouns. The ControlNet takes the semantic map merged from the masks as the layout condition, and BoxDiff takes the bounding boxes with labeled entity nouns as the layout condition.

To compare the image retrieval performance, we retrieved the target image for its corresponding reference image across all the 100 reference images that have the same reference event class.

| Model | CLIP-I↑ | DINO↑ | DreamSim↑ | CLIP-T↑ |
|---|---|---|---|---|
| **FreeEvent** | **0.5771** | **0.2865** | **0.3877** | **0.3445** |
| FreeEvent w/o guidance | 0.5140 | 0.1058 | 0.2079 | 0.2979 |
| FreeEvent w/o regulation | 0.5459 | 0.1297 | 0.3011 | 0.3205 |
| FreeEvent w/o injection | 0.4945 | 0.0825 | 0.2694 | 0.3311 |

Table 3: Quantitative results for ablation study of two paths on Real-Event.

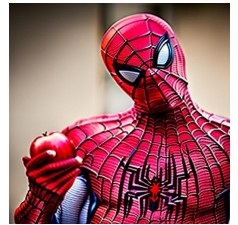
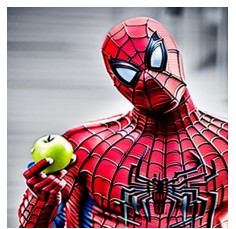
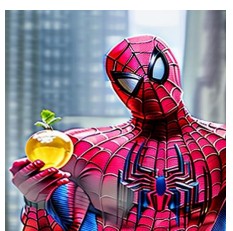

① Spiderman ② red apple       ① Spiderman ② green apple       ① Spiderman ② crystal apple

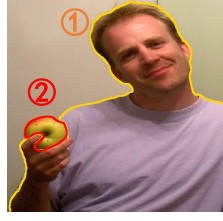

Reference Image

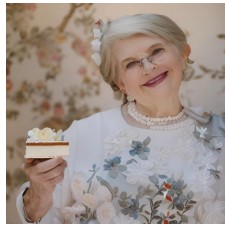
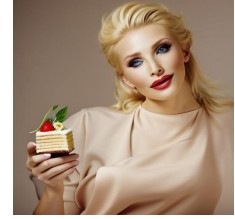
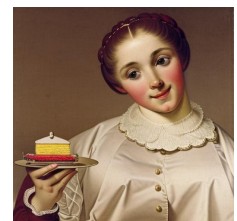

① old lady ② cake       ① blonde lady ② cake       ① noble lady ② cake

Figure 8: The results of attribute generation during event customization.

## C. Quantitative results for Path Effectiveness.

We ran ablations of two paths on Real-Event. We evaluated the CLIP-I, DreamSim, and DINO scores for image similarity between reference images and the generated images. We also evaluated the CLIP-T score for image-text similarity between text prompts and the generated images. Table 3 demonstrates the effectiveness of two paths.

## D. Attribute Generation Results.

In this paper, we didn't explicitly model the attributes during generation. However, as the results are shown in Figure 8, since we can generate extra content for background and style by giving corresponding text descriptions, we thus tried to model the attributes by giving extra adjectives to the target prompt as an easy and natural exploration. Meanwhile, to ensure the accurate generation of the attributes, we applied the cross-attention guidance and regulation on each attribute using the mask of the entity they describe. As the results shown in Figure 8, our method successfully addresses the attributes of the corresponding entity (*e.g.*, colors, materials, and ages). After all, while the attribute part is not the primary focus of our work, our approach shows potential and effectiveness in addressing it, and we would be happy to conduct further research in our future work.

## E. More Event-Subject Customization Comparisons.

In this section, we provide more event-subject customization comparisons with exisitng subject swapping and multi-subject customization methods, including Anydoor (Chen et al., 2024c), PhotoSwap (Gu et al., 2023) and MS-Diffuison (Wang et al., 2025). As shown in Figure 9, FreeEvent outperforms all other methods.

We need to clarify that the primary focus of this paper is event customization, while event-subject combined customization is only a potential capability of FreeEvent, rather than a key aspect we intend to emphasize or compare with existing methods. Moreover, FreeEvent serves as a plug-and-play framework for event-subject combined customization, making it unsuitable

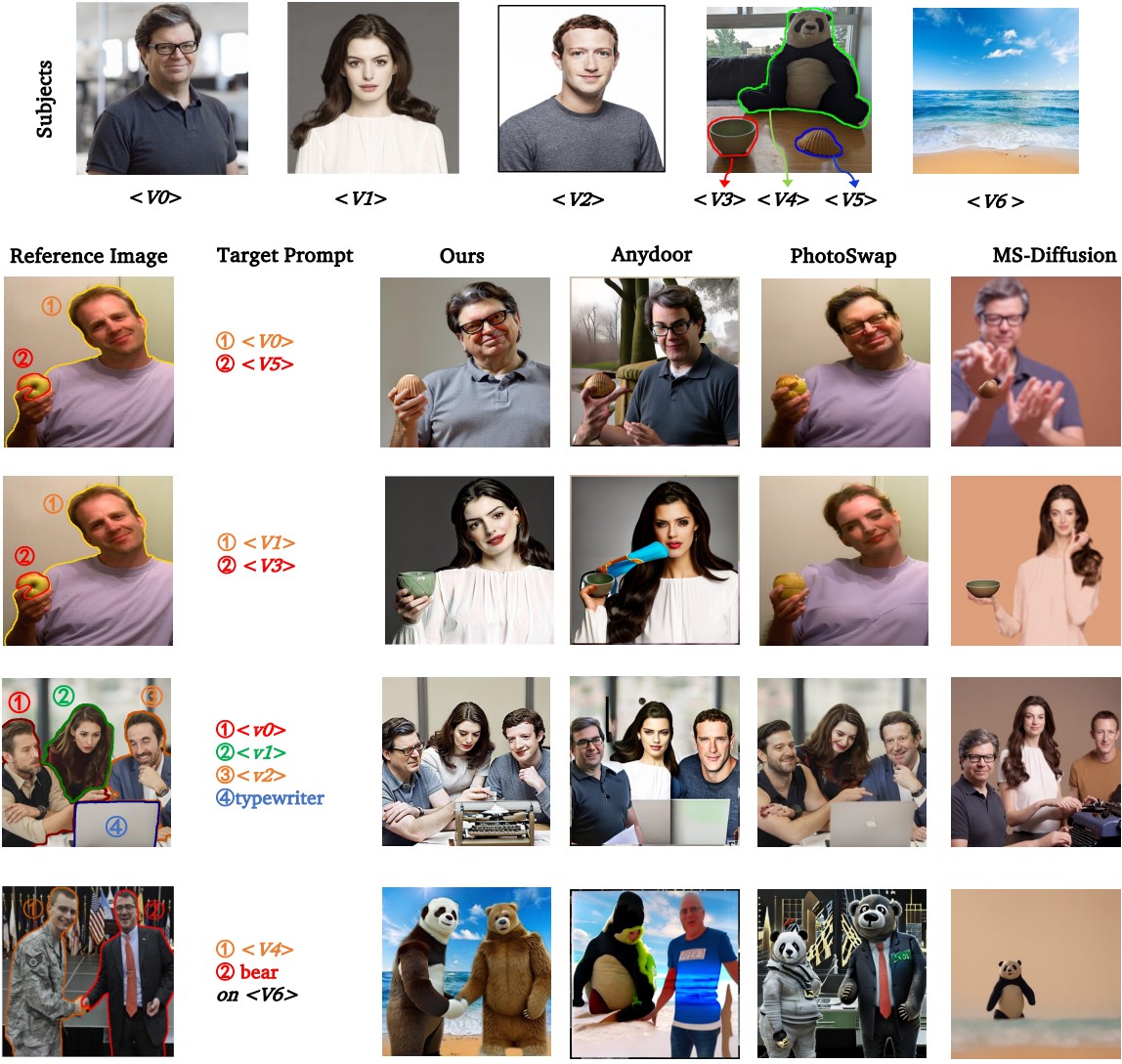

Figure 9: **Comparisions of Event-Subject Customization.** Different colors and numbers show the associations between reference entities and their corresponding target prompts.

for direct compare with subject customization methods as their settings and applicable scenarios differ. We will provide further discussions and results on event-subject customization in future work.

## F. Discussion about hyperparameter Setting.

As the first work on event customization, our goal is to enable FreeEvent to perform high-quality event customization across diverse reference images using a unified set of hyperparameters (see Appendix Sec A). This default setting ensures faithful event transferring while allowing flexibility in target entity generation.

However, when there is a large shape discrepancy between the target entity and the reference entity, the layout information from the reference entity may undesirably affect the appearance of the target entity. This reflects a fundamental trade-off between event transferring and entity switching: prioritizing accurate event customization based on the reference image may lead to some compromise in the generation of the target entity. As the example shown in Figure 10, when the shape differences are significant (horse vs. dinosaur), the default setup may result in suboptimal generation (Target Image 1).

A straightforward solution is to adjust the parameters of the event transferring and entity switching paths. Specifically,

| **Reference Image** | **Target Prompt** | **Target Image1** | **Target Image2** |
|:---:|:---:|:---:|:---:|
| 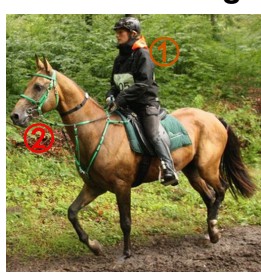 | 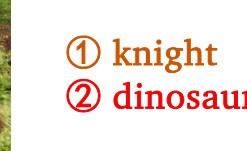 ① knight ② dinosaur | 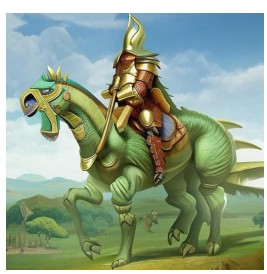 | |

|  | Cross-attention Guidance | Cross-attention Regulation | Spatial Feature Injection | Self-attention Injection |
|:---:|:---:|:---:|:---:|:---:|
| **Target Image1** | 10 steps | 50 steps | 50 steps | 25 steps |
| **Target Image2** | 15 steps | 50 steps | 30 steps | 15 steps |

Figure 10: **Comparisions of Event Customization with different hyperparameters settings.** Different colors and numbers show the associations between reference entities and their corresponding target prompts.

enhancing the entity switching to emphasize the generation of the dinosaur, while slightly reducing the strength of event transferring to mitigate layout constraints from the reference entity.

As the shown in Figure 10, we provide an updated result (Target Image 2) by increasing the number of cross-attention guidance steps (from 10 to 15) and reducing the number of injection steps to 60% of the original. This rebalancing enables a more suitable trade-off for this case, resulting in more prominent dinosaur features (*e.g.*, shorter front claws and upright back legs) while still preserving the core event structure.

This case study also demonstrates a practical and accessible way for users to adjust the trade-off between event transferring and entity switching according to their own customization needs. Looking ahead, we plan to explore more flexible and general solutions, such as adaptive parameter scheduling during generation or more explicit entity switching mechanisms, to further improve the controllability and diversity of target entities while maintaining event fidelity.

## G. Limitation.

The main limitation of FreeEvent lies in the complexity of events and the number of entities. The customization effect may be compromised when there are too many entities in an image, especially if they are too small. As the first work in this direction, we hope our method can unveil new possibilities for more complex customization and the generation of a greater number of richer, more diverse entities. Additionally, since our model is built on pretrained Stable Diffusion (SD) models, our performance depends on the generative capabilities of SD. This can lead to suboptimal results for entities that the current SD struggles with, such as human faces and hands.

## H. More Qualitative Comparision Results.

We show more comparisons on Real-Event in Figure 11, Figure 12, Figure 13, Figure 14 and Figure 15. Specifically, we list them by the order of entity numbers. And we use different combinations of target entities for the same reference image to generate diverse target images.

(The figures are in the next pages.)

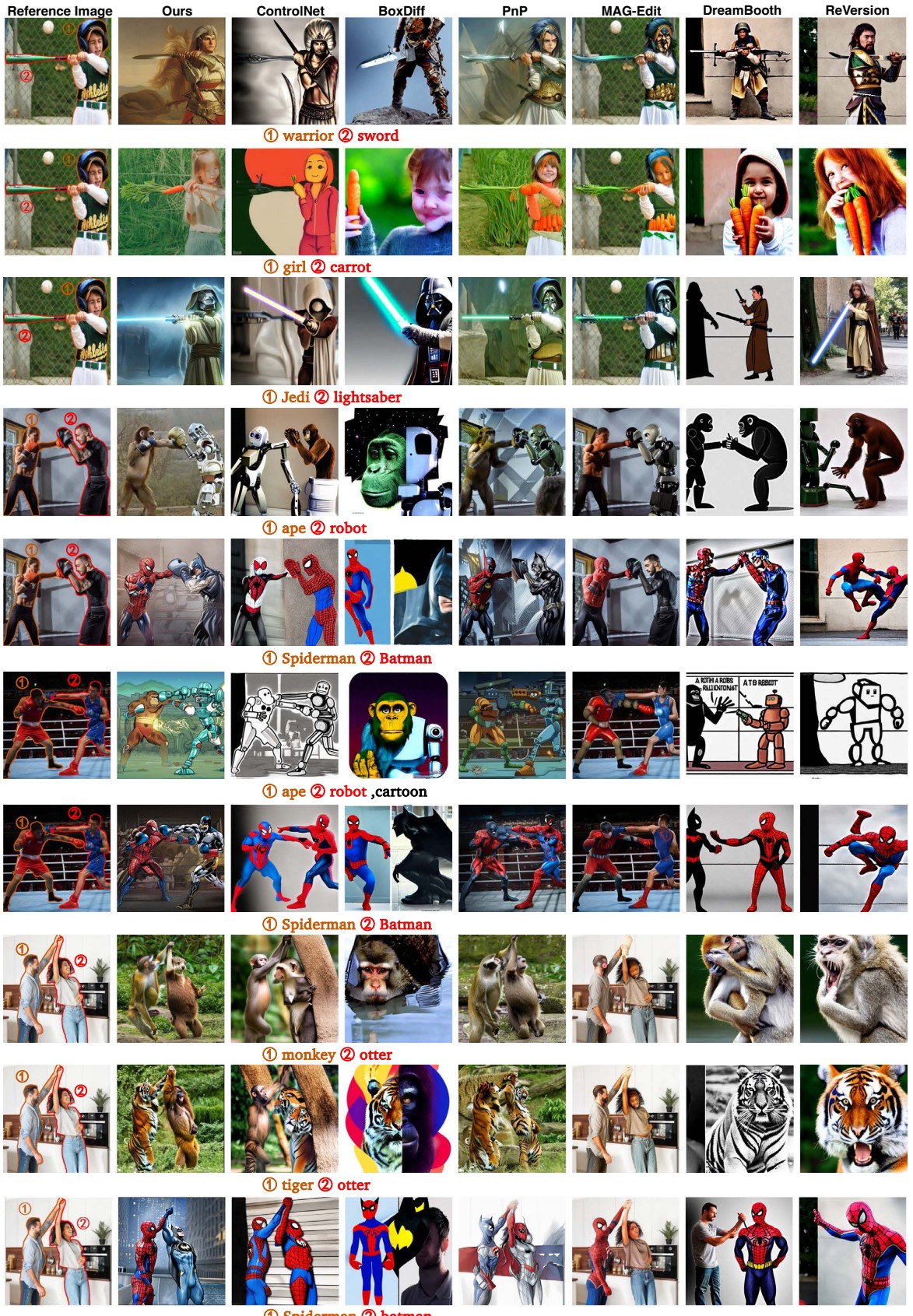

Figure 11: **Comparision of Event Customization.** Different colors and numbers show the associations between reference entities and their corresponding target prompts.

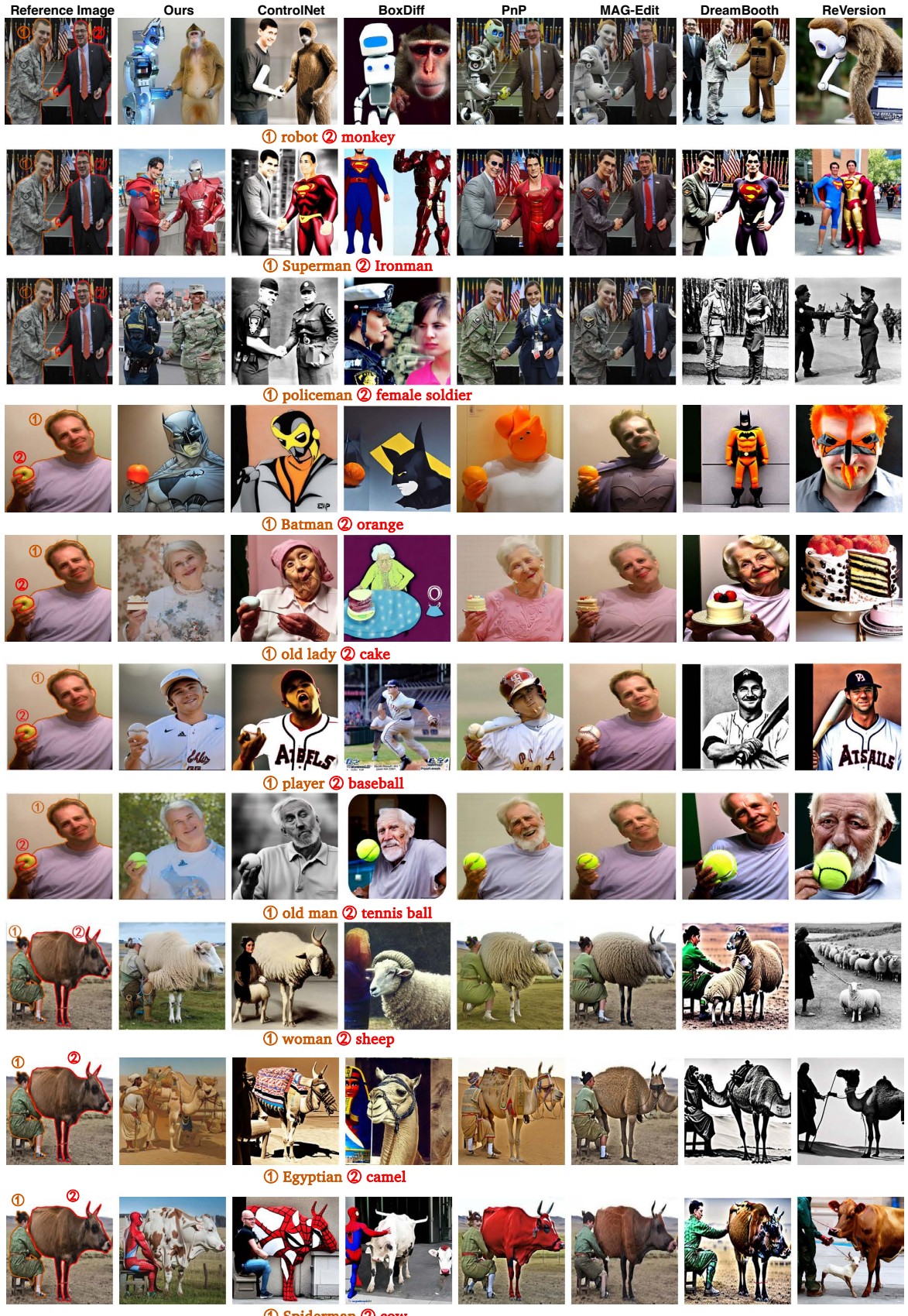

Figure 12: **Comparision of Event Customization.** Different colors and numbers show the associations between reference entities and their corresponding target prompts.

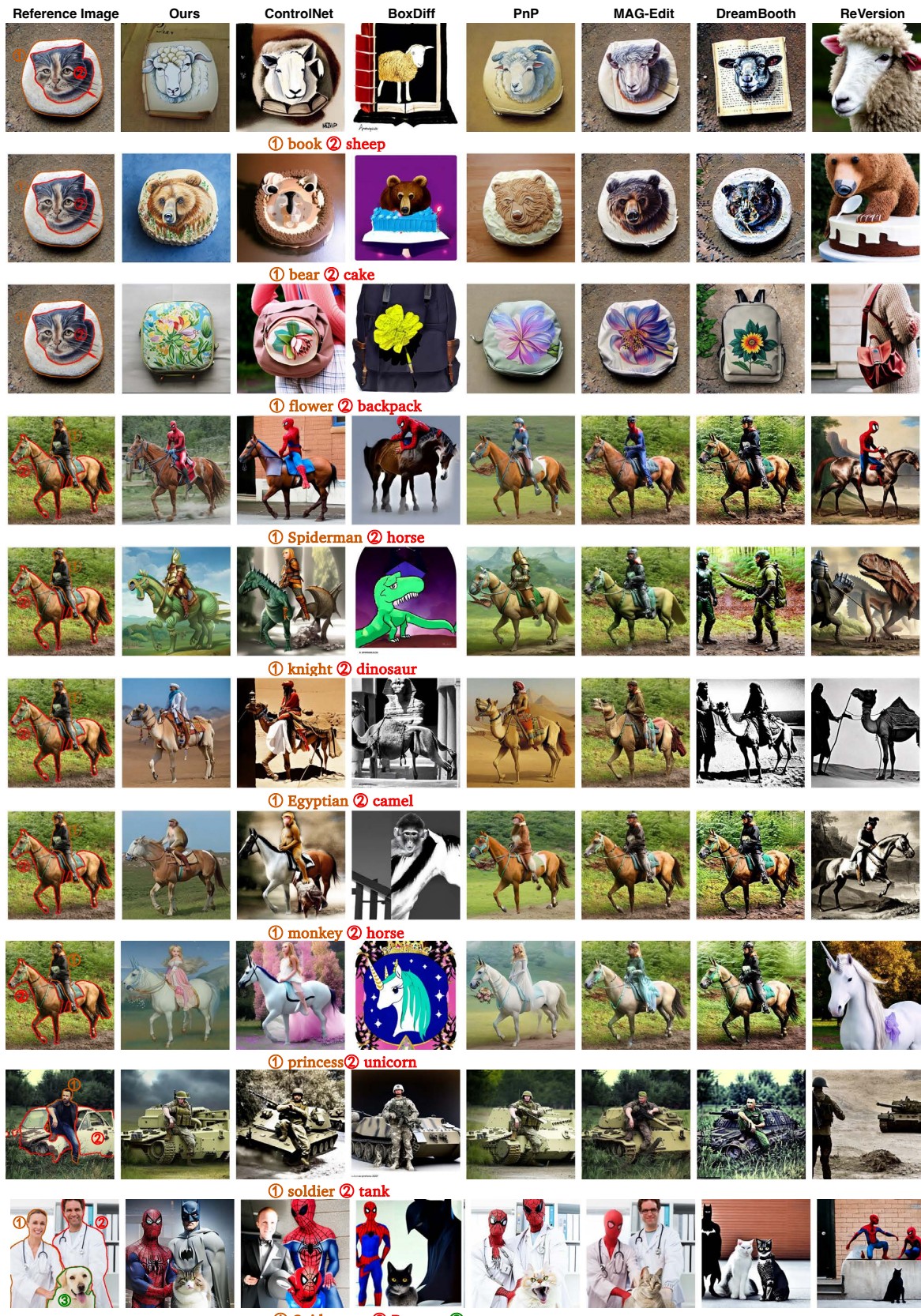

Figure 13: **Comparision of Event Customization.** Different colors and numbers show the associations between reference entities and their corresponding target prompts.

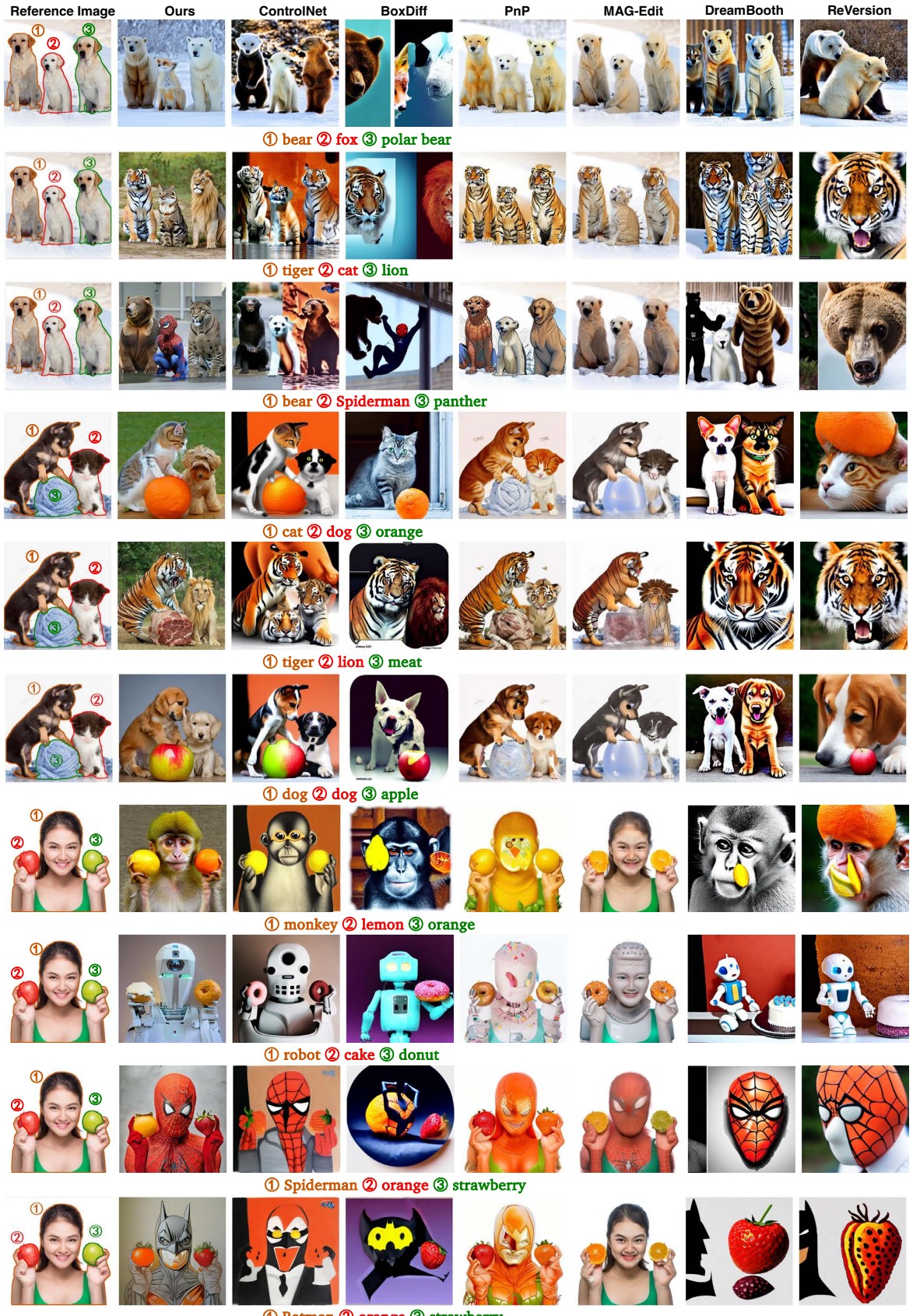

Figure 14: **Comparision of Event Customization.** Different colors and numbers show the associations between reference entities and their corresponding target prompts.

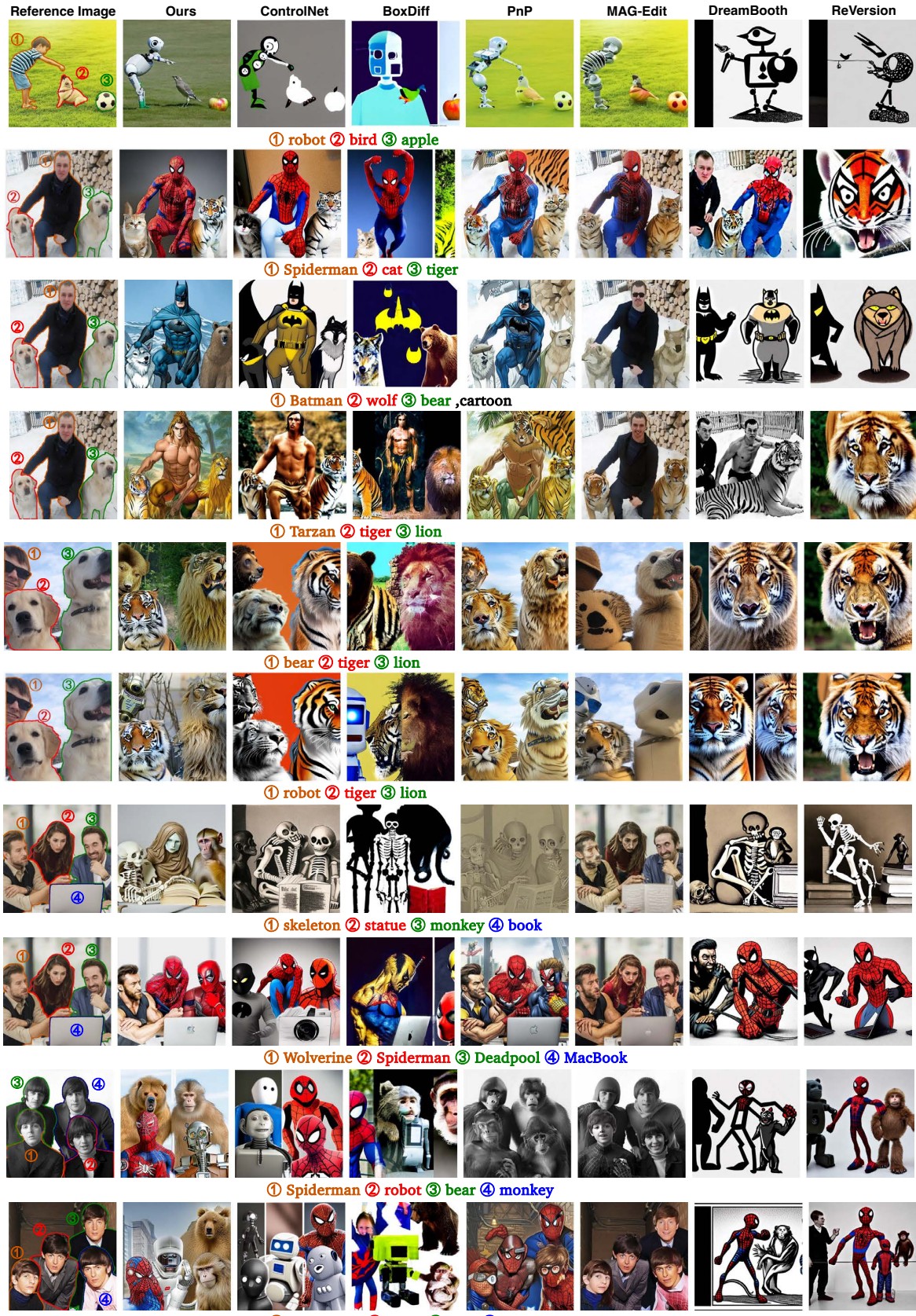

Figure 15: **Comparision of Event Customization.** Different colors and numbers show the associations between reference entities and their corresponding target prompts.

