# OpenReview forum: "Event-Customized Image Generation"
_ICML.cc/2025/Conference — ICML 2025 poster_

### Official Review · Reviewer_SWq5 · 2025-03-03

**Overall Recommendation:** 4

**Summary:**

This paper introduces event-customized image generation, a new task that extends customized image generation to complex scenes which includes detailed actions, poses, relationships, and interactions between entities. To tackle this new task, it proposes FreeEvent, a training-free method that enhances the diffusion denoising process through two key paths: the entity switching path for precise entity control, and the event transferring path to guide event generation. It further collects two benchmarks for this task. Experimental results on the two benchmarks demonstrate the advantage of FreeEvent.

**Claims And Evidence:**

This paper introduces a new task, accompanied with two datasets and a newly proposed method. However, more details on the datasets should be provided. For example, authors should provide some necessary statistics on the two datasets to better prove the coverage and diversity, and describe how the two newly collected datasets differ from their original counterparts.

**Essential References Not Discussed:**

The reference information seems adequate.

**Experimental Designs Or Analyses:**

The experimental designs and analyses seem to be comprehensive and of good soundness and validity.

**Methods And Evaluation Criteria:**

The methodology includes an entity switching path to guide the target entity generation and an event transferring path for event generation. The design appears to be reasonable. Regarding to the evaluation criteria, the authors use both qualitative and quantitative measurements such as CLIP scores. Besides that, they also employ subjective user study. The evaluation appears to be comprehensive.

**Other Comments Or Suggestions:**

n/a

**Other Strengths And Weaknesses:**

Strength:

1 This paper is well-written and easy to follow.

2 The motivation for proposing the Event-Customized Image Generation task is strong and clear, with a thorough analysis of the limitations of existing methods. The new benchmarks also enhanced the foundation of this research.

3 The proposed FreeEvent is well-designed by decomposing event generation into two key components, ensuring a reasonable and effective solution. As a training-free approach, it is also computationally efficient for real-world applications.

4 The FreeEvent demonstrated its effectiveness through extensive experiments over various baseline methods. Moreover, it produced several inspirational results, such as its combination with subject customization and background images. These findings further highlight the potential of the proposed task and method for broader applications.

Weakness:

1 It seems that the target prompt only contains the entities, what about adding the prompt of the ``event”? For example, using the event class to describe the event more detailly, will this lead to better results? The author should provide more analysis and clarifications.

2 The setting of the user study seems somewhat unreasonable. As the authors state, “every expert was asked to choose three targets.” However, based on the visualization results presented in the paper, many samples do not contain three clearly preferable images. This setting may weaken the credibility of the results. The authors should consider improving the evaluation or statistical method to more accurately reflect the user study’s findings.

**Questions For Authors:**

The paper claims that “each reference image in SWiG-Event contains 1 to 4 entities”. While SWiG-Event ensures diversity in event types (i.e., 50 kinds), what is the specific distribution of entity numbers in SWiG-Event? Is it evenly distributed to fairly evaluate the model’s capability?

**Relation To Broader Scientific Literature:**

This paper proposes the task of event customized image generation, which considers the control factors in image generation task more comprehensively than any other related studies. It paves a new way towards the image generation by the control of more complicated semantics and knowledge.

**Theoretical Claims:**

There is no theoretical claim or proofs. However, the model designs seem reasonable.

---

> ### Author Rebuttal · Authors · 2025-03-31
>
> ## Link for additional results: https://anonymous.4open.science/r/FreeEvent-EB1D/README.md
>
>
>
>
>
> ## Q1: Adding prompt for event
>
>
> **A1:** We have analyzed the impact of incorporating an explicit “event” description in the ablation studies (Appendix, Section C). Specifically, as shown in Figure 8, adding verb descriptions to the prompt negatively affects the appearance of entities in the generated images. The primary reason is that these descriptions may not align well with the pretrained diffusion model. Moreover, for practical application, accurately describing events in complex scenes can be challenging for users. Since our method already enables precise extraction and transfer of reference events, users are not required to explicitly specify events in the target prompt. This further highlights the practicality of FreeEvent.
>
> Apart from verb description, we also analyzed the influence of adding the background, style and attribute description. As shown in Section C and Section E, our method can accurately generate these additional elements without affecting the event. Theses results all demonstrates FreeEvent’s strong generalization capability.
>
>
>
> ## Q2: Setting of the user study
>
>
> **A2:** Thanks for the suggestion. We have adjusted the setting of “choose three targets” to “select at least one and up to three” and conducted a new user study. This updated setting better reflects expert judgment and highlights the differences between different methods. We also included two more baselines suggested by other reviewers. Table R3 in the link shows that our approach achieves the best performance in human judgments (HJ).
>
>
>
>
> ## Q3: Distribution of SWiG-Event
>
> **A3:** The distribution of entity numbers (1, 2, 3, 4) in SWiG-Event is (20%, 30%, 30%, 20%). As the first benchmark for event customization, we aimed for an overall balanced distribution while making slight adjustments—specifically, increasing the proportion of two- and three-entity cases. This decision ensures that the benchmark remains well-rounded: single-entity events may be too simplistic, while four-entity events could be overly challenging. By striking this balance, SWiG-Event serves as a well-designed benchmark for effectively evaluating model capability.

---

### Official Review · Reviewer_SH2p · 2025-03-13

**Overall Recommendation:** 3

**Summary:**

This paper introduces a new task called event customization, which aims to generate new images that maintain the same event depicted in a reference image. An event contains specific actions, poses, relationships, and interactions between different entities within a scene. To address this task, a training-free FreeEvent is proposed that integrates an entity switching path and an event transferring path into the diffusion denoising process. Additionally, two benchmarks are proposed to evaluate the proposed method both qualitatively and quantitatively.

**Claims And Evidence:**

The claims are clear and easy to understand.

**Essential References Not Discussed:**

N/A

**Experimental Designs Or Analyses:**

Extensive experiments have been conducted. However, there are some concerns:

1. In Table 1, only DreamBooth and BoxDiff are compared. It is recommended to include additional methods to ensure a more comprehensive evaluation.
2. It would be beneficial to compare with text/reference-based image inpainting tasks, as these methods can also generate target concepts in specified regions.

**Methods And Evaluation Criteria:**

The FreeEvent is reasonable. The proposed benchmarks are diverse and adequate for evaluation, and the metrics used are suitable.

**Other Comments Or Suggestions:**

N/A

**Other Strengths And Weaknesses:**

Strengths
1. The proposed event customization task is interesting and new.
2. The proposed method is straightforward and easy to follow.

Weaknesses

1. The novelty may be limited, as discussed in the "Relation to Broader Scientific Literature" section.
2. In the figures in the paper, the target concepts usually share a similar shape to the one in the reference image. How well does the proposed method perform if they have different shapes, for example, a human -> a tiger?

**Questions For Authors:**

My main concerns have been listed in the weaknesses part.

**Relation To Broader Scientific Literature:**

While the task of event customization is new, the key techniques have been explored in recent methods. For example, the entity switching path is similar to that in [1], and the event transferring path resembles the approach used in [2].

[1] Chefer, Hila, et al. "Attend-and-excite: Attention-based semantic guidance for text-to-image diffusion models."
[2] Cao, Mingdeng, et al. "Masactrl: Tuning-free mutual self-attention control for consistent image synthesis and editing."

**Theoretical Claims:**

N/A

---

> ### Author Rebuttal · Authors · 2025-03-31
>
> ## Link for additional results: https://anonymous.4open.science/r/FreeEvent-EB1D/README.md
>
>
>
>
>
> ## Q1: Additional baseline methods
>
>
> **A1:** Considering both reviewer yvsP and t7sW’s comments and the limited rebuttal time, we have incorporated a more recent baselines MIGC (2024) for a more comprehensive comparison. As shown in Table R1,R3 and Figure R1 in the link, our FreeEvent succesfully outperforms it.
>
>
>
>
> ## Q2: Compare with image inpainting methods
>
>
> **A2:** We compared our method with the text-based image inpainting approach Blended Latent Diffusion on Real-Event. As shown in Table R3 and Figure R1 in the link, FreeEvent consistently outperformed it.
>
> While image inpainting methods can generate target entities in specified regions, they primarily focus on independently adding or replacing objects at the entity level, often overlooking interactions between them. As a result, directly using masks and text prompts for inpainting can disrupt the event details in the reference image, altering entity poses, actions, and interactions.
>
> Additionally, Sec. 4.3 presented a comparison with MAG-Edit, a localized image editing method that utilizes reference entity masks for region-specific modifications. Our results demonstrated that FreeEvent also surpasses MAG-Edit in preserving event details and interactions.
>
>
> ## Q3: About Novelty
>
>
> **A3:** Thanks for your concerns. We want to first emphasize that we made three folds of contributions in this paper: 1) The new and meaningful event-customized image generation task. 2) The first training-free method for event customization. 3) Two evaluation benchmarks for event-customized image generation. Specifically, for our training-free method FreeEvent, we provide more discussion below.
>
> - **Motivation.** Based on the two main components of the reference image, i.e., entity and event, we proposed to decompose the event customization into two parts: 1) Switching the entities in the reference image to target entities. 2) Transferring the event from the reference image to the target image. Inspired by the observation that the spatial features and attention maps have been utilized to control the layout, structure, and appearance in text-to-image generation, we further designed the two corresponding paths to address the two parts. While these observations have been widely recognized in previous works, we are the first to integrate them to address this new task in a training-free manner. This approach demonstrates a thoughtful analysis of the task and a strategic application of existing technologies.
>
> - **Improvements.** We also made several specific improvements to better address the event customization task. 1) For entity switching, besides the cross-attention guidance, we further regulate the cross-attention map of each entity to avoid the appearance leakage between each target entity. 2) For event transferring, in contrast to previous works [A, B] that perform DDIM inversion on reference images, we directly use forward diffusion. This further reduces the appearance leakage from the reference image and saves the inversion cost and additional model inference time.
>
> While FreeEvent does incorporate some existing methods, its design is rooted in a thoughtful analysis of the new task and a strategic application of existing insights. Furthermore, we also introduced specific improvements, enabling it to address this new task more effectively and efficiently. FreeEvent has proved its effectiveness and efficiency in a wide range of experiments, beating existing controllable generation, image editing, and customization works. As the first work in this direction, we hope our method can unveil new possibilities for more complex customization, meanwhile serving as a challenging baseline for future works.
>
> [A] Plug-and-play diffusion features for text-driven image-to-image translation. CVPR, 2023.
>
> [B] Masactrl: Tuning-free mutual self-attention control for consistent image synthesis and editing. ICCV, 2023.
>
>
>
> ## Q4: Reference and target concepts with different shape
>
>
> **A4:** Many samples in our paper show the performance of FreeEvent when handling target concepts that has different shape with the one in reference image.
>
> 1) Figure 4, row 2, dog -> bird.
> 2) Figure 4, row 4, dog -> Spiderman.
> 3) Figure 4, row 6, human -> bear.
> 4) Figure 11, row 8, woman -> otter.
> 5) Figure 11, row 9, man -> tiger.
> 6) Figure 13, row 7, human -> monkey.
> 7) Figure 15, row 6, human -> robot.
>
> As the above results shown, ControlNet and the localized image editing models tend to generate the target entities strictly matching the shape of their corresponding reference entities, which appears incongruous. On the contrary, the entities generated by FreeEvent not only match the layout of the reference entity but also keep it harmonious with different shapes, allowing for more diverse generation of target entities.

---

> > ### Comment · Reviewer_SH2p · 2025-04-04
> >
> > Thank you to the authors for their efforts. After reading the rebuttal, most of my concerns have been addressed. Although I still believe the technical contribution is not very significant, the event customization is interesting and valuable for the application. Therefore, I will raise my rating to 3.
> >
> > Besides, I have another concern. As mentioned by Reviewer yvsP and the authors, an "event" is defined as all specific actions, poses, relations, or interactions between different entities in the scene. From the figures presented in the paper, the poses seem to replicate the layout from the reference image. In Fig. 13, row 5, the generated dinosaur's layout is identical to that of the reference horse. Since the shape of a dinosaur differs from that of a horse, for example, the dinosaur has shorter front claws, this results in the generated output appearing unrealistic. It would be beneficial to address these discrepancies to enhance the method's applicability.

---

> > > ### Author Response · Authors · 2025-04-04
> > >
> > > We sincerely appreciate your feedback and decision.
> > >
> > > Regarding the concern you just raised, we have indeed encountered similar issues in our experiments. Specifically, when there is a large shape discrepancy between the target entity and the reference entity, the layout information from the reference entity may undesirably affect the appearance of the target entity. This reflects a fundamental trade-off between event transferring and entity switching: prioritizing accurate event customization based on the reference image may lead to some compromise in the generation of the target entity.
> > >
> > > As the first work on event customization, our goal is to enable FreeEvent to perform high-quality event customization across diverse reference images using a unified set of hyperparameters (see Appendix Sec A). This default setting ensures faithful event transferring while allowing flexibility in target entity generation. However, as the mentioned example in Fig. 13 (row 5), when the shape differences are significant (horse vs. dinosaur), the default setup may result in suboptimal generation.
> > >
> > > A straightforward solution is to adjust the parameters of the event transferring and entity switching paths. Specifically, enhancing the entity switching to emphasize the generation of the dinosaur, while slightly reducing the strength of event transferring to mitigate layout constraints from the reference entity.
> > >
> > > We have updated Figure R5 in link, we provide an updated result by increasing the number of cross-attention guidance steps (from 10 to 15) and reducing the number of injection steps to 60% of the original. This rebalancing enables a more suitable trade-off for this case, resulting in more prominent dinosaur features (e.g., shorter front claws and upright back legs) while still preserving the core event structure.
> > >
> > > This case study also demonstrates a practical and accessible way for users to adjust the trade-off between event transferring and entity switching according to their own customization needs.
> > >
> > > Looking ahead, we plan to explore more flexible and general solutions, such as adaptive parameter scheduling during generation or more explicit entity switching mechanisms, to further improve the controllability and diversity of target entities while maintaining event fidelity.
> > >
> > > Again, thank you for your thoughtful feedback. If you have any further questions or suggestions, please feel free to let us know.

---

### Official Review · Reviewer_t7sW · 2025-03-17

**Overall Recommendation:** 3

**Summary:**

In the area of customized image generation, existing methods face the limitations of simplified customizations and insufficient data. To address these challenges, this paper defines a novel task, event-customized image generation, covering complex layout, actions, interactions between more than two objects. The training free method consists of two paths: the entity switching path and the event transferring path, via manipulation of cross attention, self attention and spatial features. The authors conclude that the proposed method can be a plug-and-play module for other models and is able to support more complex scenes. They also collect two benchmarks for the evaluation of this new task.

**Claims And Evidence:**

- In the teaser, abstract and introduction, the authors first list the tasks of subject customization and action customization, and then indicate that the proposed task/method can address the challenges of the older tasks. However, in most part of the paper, subject customization is not mentioned (except that in Fig. 6 it is briefly discussed), and FreeEvent seems more like a pose/layout-guided class-conditioned generation. The story needs to be refined by either removing the subject customization or improving FreeEvent to naturally support subject personalization.
- In introduction, the authors claim that "gathering images that depict the exact same action or interaction is challenging", which is a very reasonable constraint for the previous method to learn a specific action.
- At the end of Sec. 3.3, the paper claims that the "framework can be easily combined with subject customization methods"; however, it is limited to UNet architectures (based on SD v2.1), so it is questionable whether this method can be extended to DiT-based methods (e.g., FLUX, SD3, etc.).

**Essential References Not Discussed:**

Please see the methods mentioned in "Methods And Evaluation Criteria".

**Experimental Designs Or Analyses:**

- The issues of the experiments mainly lie in the metrics design and baseline selection. Please refer to "Methods And Evaluation Criteria" above.
- To demonstrate that the proposed method can be easily integrated into other models, there should be more discussion on injecting FreeEvent into other models, such as SD-XL.
- The current ablation study only shows visual results (Fig. 5), which cannot reflect the general performance. More quantitative results will. help.

**Methods And Evaluation Criteria:**

- In Tab. 1, to evaluate the image alignment of the generated images with the references, CLIP-I is used. Why not use other image-based metrics such as DINO score and DreamSim?
- To demonstrate the advancement of the proposed model, more layout-guided T2I methods should be included for comparison, such as GLIGEN and LayoutDiffusion (Tab. 1).
- In this new task, current metrics cannot effectively measure the accuracy of the poses/interactions. Better metrics should be designed for this task to evaluate the layout/pose preservation (e.g., AP).
- Subject customization is ignored in most part of the paper. There should be more discussion and results shown other than Fig. 6 and Fig. 10: 1) comparison with existing methods, such as PhotoSwap and MS-Diffusion; 2) use reference images with more complex textures, such as DreamBench.

**Other Comments Or Suggestions:**

No other comments.

**Other Strengths And Weaknesses:**

- Although the chosen metrics cannot fully reflect the effectiveness of the methods, the paper includes a user study in Sec. 4.5, where human preference demonstrate the quality of FreeEvent.
- The method section is well written and easy to understand.
- The left part of Fig. 2a is a little hard to understand (the noise maps), may need to be revised.

**Questions For Authors:**

- Does the proposed method work for architectures other than UNet? E.g., DiT?

**Relation To Broader Scientific Literature:**

In previous papers of image customization or conditioned T2I, the use case of replacing the entities from an image with specific classes/concepts is usually overlooked. In response, this paper defines a new task for this missing scenario.

**Theoretical Claims:**

Equation 3), 4), 5) look correct and reasonable to me.

---

> ### Author Rebuttal · Authors · 2025-03-31
>
> ## Link for additional results: https://anonymous.4open.science/r/FreeEvent-EB1D/README.md
>
>
>
>
>
> ## Q1: About story
>
>
> **A1:** Our intention in introducing subject customization and action customization is to provide background on the broader customization task and naturally introduce event customization. Our main focus is event customization, while event-subject combined customization is only a potential capability of FreeEvent rather than a key aspect we aim to emphasize or compare with existing methods. We acknowledge your concern and will refine our writing to clarify the relationships between these tasks, ensuring a clearer distinction between FreeEvent’s primary capabilities and its potential abilities.
>
>
> ## Q2: More baselines and metrics
>
> **A2:** Since GLIGEN and LayoutDiffusion were proposed in 2023, considering reviewer yvsP’s comments and limited rebuttal time, we incorporated a more recent layout-guided baselines MIGC. Additionally, we reported the DINO score and DreamSim. As shown in Table R1,R3 and Figure R1 in the link, FreeEvent successfully outperforms all methods.
>
>
>
>
>
> ## Q3: Design better metrics
>
> **A3:** We need to first emphasize that evaluating the accuracy of complex events is a challenging and open problem. Similar to other customization tasks (e.g., subject or action customization), existing works also face evaluation difficulties and primarily rely on similarity metrics or user studies. Moreover, designing new metrics requires extensive effort, experimentation, and validation, which go beyond the scope of this work. Regarding metrics like AP, they are not suitable in our case, as we do not impose strict constraints on the exact shape or layout of the generated target entities at pixel level.
>
> In our experiments, we tried to evaluate event customization from multiple perspectives: 1) Global image similarity: retrieval results and CLIP-I score. 2) Entity similarity: CLIP-T score. 3) Event similarity: verb detection results. Together with user study, we believe these metrics provide a comprehensive evaluation of the event customization quality. As the first work for this direction, we hope our evaluation and metrics serve as a valuable starting point, and we will explore more suitable metrics in future work.
>
>
> ## Q4: About subject customization
>
>
> **A4:** We provided more event-subject customization comparisons in Figure R2 in the link, and FreeEvent outperforms all other methods.
>
> However, we need to clarify that the primary focus of this paper is **event customization**, while **event-subject combined customization** is only a potential capability of FreeEvent, rather than a key aspect we intend to emphasize or compare with existing methods. Moreover, FreeEvent serves as a plug-and-play framework for event-subject combined customization, making it unsuitable for direct compare with subject customization methods as their settings and applicable scenarios differ.
>
> As mentioned in Q1, we will refine our writing to better clarify the relationships between these tasks and provide further discussions and results on event-subject customization in future work.
>
>
> ## Q5: Quantitative results for ablation study
>
>
> **A5:** We ran ablations of two paths on Real-Event. We evaluated the image and text similarities with reference image and text prompt, respectively. Table R2 in the link demonstrates the effectiveness of two paths.
>
>
> ## Q6: Choice of layers
>
>
> **A6:** Our choices follow widely acknowledged empirical insights from diffusion models:
>
> 1) During object generation, early denoising steps determine layout and position, while later steps refine appearance.
>
> So we only apply attention guidance in early steps, which also saves the inference time.
>
> 2) Spatial features and attention maps from decoder layers encode localized layout information.
> 3) Injecting features in deeper layers can better preserve the structure but risk appearance leakage.
>
> So we only inject spatial features at first decoder layer but attention maps at all decoder blocks.
>
>
>
>
> ## Q7: Does FreeEvent works with other models
>
>
>
> **A7:** For Unet-based models, the insights from Q6 can be easily transfered, making FreeEvent easily integrable into SD-XL. Specifically, for 50-step DDIM sampling in SD-XL v1.0: 1) spatial feature injection: {decoder block 1}. 2) Self-attention injection: {decoder block 1,2,3}. 3) cross-attention guidance in first 10 steps. Figure R3 in the link shows various results.
>
> For DiT-based models, due to their distinct architecture from UNet, direct injection of FreeEvent is challenging. However, the core ideas of entity switching and event transferring can still be adapted using similar insights:
> 1) Leveraging cross-attention on text tokens to guide target object generation.
> 2) Modifying attention on visual tokens to control structure generation.
>
> As the first work in this direction, we hope FreeEvent serves as an effective benchmark and look forward to exploring its potential on DiT models in future work.

---

> > ### Comment · Reviewer_t7sW · 2025-04-09
> >
> > I appreciate the responses and the additional experiments from the authors. Based on these results, I'm willing to increase my rating.

---

> > > ### Author Response · Authors · 2025-04-09
> > >
> > > Thank you for increasing the rating. Your valuable suggestions greatly contribute to the quality of our manuscript. Thank you again for your precious time and thoughtful feedback!

---

### Official Review · Reviewer_yvsP · 2025-03-24

**Overall Recommendation:** 3

**Summary:**

This paper introduces FreeEvent, a diffusion-based image generation technique designed to address the Event-Customized image synthesis problem identified in this study. The authors define this problem by analyzing the progress and limitations of existing controllable image generation methods, particularly highlighting two key challenges: (a) overly simplified customization and (b) the reliance on multiple reference images, which is especially impractical in event-customized generation scenarios.

To overcome these challenges, FreeEvent incorporates two novel pathways in addition to the standard diffusion denoising process: the Entity Switching Path and the Event Transferring Path. Experimental results on two benchmark datasets demonstrate the effectiveness of FreeEvent, showcasing its superiority over existing methods. Furthermore, an ablation study confirms the contributions of the two proposed pathways to the overall performance.

**Claims And Evidence:**

The experimental results validate the capability of the proposed FreeEvent in customizing the identities of multiple instances in the reference image, as well as their interactions, which are represented by attributes such as pose, action, and relationships. However, whether such customization of identities and interactions fully aligns with the concept of "event" remains somewhat subjective. Additionally, the adequacy of the evaluation metrics employed in the experimental section to effectively measure the quality of event customization is open to question. For further details, please refer to the following section.

**Essential References Not Discussed:**

To the best of my knowledge, there is no such an essential reference missing, although the benchmark methods are sowehow out-of-date.

**Experimental Designs Or Analyses:**

I have two primary concerns regarding the experimental designs:

1) The rationale behind the selection of evaluation metrics: As described in the "Methods and Evaluation Criteria" section, the justification for choosing the specific metrics is insufficiently detailed. A more thorough explanation is required to establish the relevance and appropriateness of these metrics in assessing the proposed method.

2) The limitations of the benchmark methods used for comparison: The methods included for benchmarking are somewhat restricted and already outdated, as most were proposed in 2023. I strongly encourage the authors to incorporate more recent and relevant methods formally published in 2024 and beyond. For instance, AnyDoor and MIGC (CVPR 2024), as well as MIGC++ (TPAMI 2025), could provide a more comprehensive and up-to-date comparison.

**Methods And Evaluation Criteria:**

This section highlights the primary concerns of this study, which will be elaborated as follows:

1) As stated in the abstract, the authors define an "event" as **ALL** specific actions, poses, relations, or interactions between different entities in the scene. How do the authors ensure that **ALL** such attributes, including the most subtle and nuanced ones, are adequately considered in the customization process? I strongly encourage the authors to provide a clearer and more measurable definition of the concept of "event" to enhance its interpretability and reproducibility.

2) Furthermore, in the "Event Transferring Path" described in Section 3.3, the definition of "event" shifts to "essentially the structural, semantic layout, and shape details of the image," which are said to be captured by "spatial features and self-attention maps" as suggested by existing studies (so convenient isn't it?). This apparent inconsistency in the definition of "event" is confusing. Earlier, the term "event" encompassed abundant attributes such as actions, poses, relations, and interactions, yet in the technical implementation, it is reduced to "structural, semantic layout, and shape details of the image." Given the elevated expectations set by the authors, this simplification raises serious concerns. Specifically: Why are "actions, poses, relations, and interactions among instances" considered equivalent to "structural, semantic layout, and shape details of the image"? Why can "structural, semantic layout, and shape details of the image" be adequately represented by spatial features and self-attention maps? These crucial questions must be explicitly addressed and thoroughly discussed to clarify the proposed method's rationale.

3) The authors employ three evaluation metrics to measure the effectiveness of the proposed method. However, it remains unclear whether these metrics can accurately reflect the capability of event customization. Instead of merely introducing the implementation details of these metrics, I encourage the authors to establish a clear connection between the chosen metrics and specific aspects of the method's performance. For example, How does the CLIP score relate to the quality of event customization? Does higher CLIP score necessarily reflect that the 'event' is better transferred? A more detailed discussion linking the evaluation metrics to the core objectives of the study would significantly strengthen the validity of the experimental results. Also, if the definition of 'event' is '**ALL** specific actions, poses, relations, or interactions', why not use more straightforward evaluation metrics to measure the alignment of these concrete aspects?

**Other Comments Or Suggestions:**

Line 407-408, doesn't -> does not

**Other Strengths And Weaknesses:**

None

**Questions For Authors:**

Please refer to Methods And Evaluation Criteria for my primary concerns regarding this work. I would like to raising my rating if my conerns are well-addressed.

**Relation To Broader Scientific Literature:**

As discussed by the authors, this study primarily addresses the following two limitations: 1) restricted and overly simplistic interactions among instances, and 2) the dependency on multiple reference images.

**Theoretical Claims:**

This is not applicable, as no novel theoretical claims have been proposed in this study.

---

> ### Author Rebuttal · Authors · 2025-03-31
>
> ## Link for additional results: https://anonymous.4open.science/r/FreeEvent-EB1D/README.md
>
>
>
>
>
> ## Q1: The defination of ``event"
>
>
> **A1:** In early NLP tasks, event was defined “an occurrence of an activity that happens at a particular time and place” [1]. Later, in visual scene analysis, works as GSR [2] concretized activities and situations by representing them through entities’ roles (nouns), locations (bounding boxes), and their associated verbs (actions, poses, interactions).
>
> In our work, considering the context and limitations of existing customization works, where current methods independently address only simple actions or interactions, we use "event" to extend these concepts to a more general setting. This allows us to cover a broader range of visual scenarios that include diverse actions, poses, relations, and interactions. At the same time, following prior research on event definitions, we maintain entity-centric settings by explicitly considering their roles (text prompts for entities) and locations (masks). Meanwhile, this also provide a straightforward and adaptable way to measure an event, i.e., by the number of entities. This is also consistent with existing methods for measuring activities and situations, which often rely on the number of involved entities and their bounding boxes.
>
>
> After all, we want to emphasize that our definition of “event” aligns with existing research while refining and specifying it within the context of customization. As the first work in this direction, our goal is to generalize customization beyond isolated actions, poses, or interactions and unify them under a broader and more structured definition. We will refine our descriptions, particularly regarding the use of terms like “ALL” in our revision.
>
>
> [1] Event Extraction: A Survey. 2022
> [2] Grounded situation recognition with transformers. 2021
>
>
> ## Q2: About Event Transferring Path
>
>
> **A2:** First, we want to clarify that this is not a shift or reduction in the definition of “event”. Instead, we are analyzing the event customization from the perspective of visual spatial information, while subject customization is more related to visual appearance. Specifically, the actions, poses, and interactions of entities in a reference image are closely tied to its spatial information. Actions and poses for each entity often correlate with fine-grained shape details (e.g., limb positioning, body orientation), while interactions between different entities are more linked to the semantic layout (e.g., the relative locations, physical contact). This perspective provides a novel approach for implementing event customization, as it allows us to leverage spatial representations to effectively capture and transfer event-related details.
>
> Besides, the usage of spatial features and self-attention maps are empirically based on common obervactions that they are highly related to image structure and layout informations. And it has also been widely acknowledged by existing diffusion works.
>
> After all, the event transferring path is designed based on the analysis of events and the reasonable application of existing findings. We will refine our descriptions in the revision to ensure clarity, and avoid terms like “essentially” to prevent potential misunderstandings.
>
>
>
>
>
> ## Q3: The evaluation metrics
>
>
> **A3:** First, directly evaluating the alignment of complex events remains a challenging and open problem. As introduced in Sec 4.2 and Appendix (Sec B), our quantitative experiments are designed to **reproduce** the reference image while maintaining the same reference event and entities. Thus, our evlautions follow the principle of **whether generated images are matched/aligned/similar with their reference images** from different perspectives: 1) Global image similarity: the retrieval results and the CLIP-I score. 2) Entity similarity: the CLIP-T score. 3) Event similarity: the verb detection results. Together, these metrics give a comprehensive evaluation of the event customization quality based on our specific settings. Additionally, we also evluated FID score to ensure the overall quality of generated images.
>
>
>
>
>
>
> ## Q4: More benchmark methods
>
>
> **A4:** Since checkpoint for MIGC++ is now not available, we incorporated MIGC for a more comprehensive quantitative comparison. We also evluated two more metrics for global image similarity suggested by reviewer t7sW (DINO score and DreamSim). As shown in Table R1,R3 and Figure R1 in the link. FreeEvent succesfully outperforms MIGC.
>
> Since Anydoor is designed for image customization, we compared it with FreeEvent on the event-subject customization setting. As shown in Figure R2 in the link. FreeEvent outperforms Anydoor and other subject customization method. Besides, we need to clarify that **event-subject combined customization** is only a potential capability of FreeEvent, rather than a key aspect we intend to emphasize or compare with existing methods.

---

> > ### Comment · Reviewer_yvsP · 2025-04-03
> >
> > I appreciate the effort made by the authors in preparing a detailed rebuttal and providing comprehensive additional results. Some of my previous concerns, specifically Q3, Q4, and part of Q2, have been largely addressed. I encourage the authors to incorporate the new materials presented in the rebuttal into the updated version of the manuscript.
> >
> > However, after carefully reviewing the rebuttal, I still have the following concerns regarding the concept of 'Event':
> >
> > 1. My original concern was: Does the concept of 'event' encompass **ALL** specific actions, poses, relationships, or interactions between different entities in the scene? Unfortunately, I still do not have a clear and direct answer to this question.
> >
> > 2. It seems the authors are eager to validate the definition of 'event' by aligning it with benchmark studies. However, I am not questioning whether the definition of 'event' is valid, as this is inherently subjective. Instead, my concern lies in understanding which visual attributes in the reference image are practically transferred by FreeEvent. To clarify, this is not about reiterating the conceptual definition of 'event,' but rather identifying the specific attributes that are preserved and transferred in practice. A clear and explicit list of these attributes would be helpful.
> >
> > 3. Based on the authors' response to Q2, I would like to understand the specific attributes or aspects in which FreeEvent extends or outperforms existing studies. Providing clear and concrete examples that demonstrate these advantages would strengthen the claims of the manuscript.

---

> > > ### Author Response · Authors · 2025-04-03
> > >
> > > Thanks for your concern. We are willing to address all the mentioned questions.
> > >
> > >
> > > ## Q1: Whether 'event' encompass **ALL** actions, poses, relationships, or interactions
> > >
> > > **A1:** Yes, the ultimate goal of event customization is to ideally encompass **ALL** actions, poses, relationships, and interactions between different entities in the scene. However, as you pointed out, ensuring that all such attributes, including the most subtle and nuanced ones are fully considered in the customization process is a critical challenge.
> > >
> > > To address this, we adopt the entity-centric setting, explicitly considering entities’ roles (text prompts) and locations (masks). The scope of an ‘event’ in customization can be then measured based on the number and location of entities. For instance, in Figure 11 (first row), the event “a kid is holding a baseball bat” is measured by the two key entities, i.e., the kid and the baseball bat, along with their masks. The 'event' of this image then encompasses the kid’s pose, the interaction between the kid and the bat, and the action of holding the bat.
> > >
> > > Furthermore, by incorporating additional entities and refining masks, the event scope can be expanded. For example, if we also consider the “baseball helmet” as an entity and apply a corresponding mask, the event would further encompass the interaction between the kid and the helmet, and allowing for more detailed customization. We have updated some examples in Figure R4 in the link.
> > >
> > > In summary, as the first work in this direction, while fully encompassing **ALL** actions, poses, relationships, and interactions remains a significant challenge, we provide a straightforward and flexible approach to measure events. By defining the main entity and applying corresponding masks, we aim to encompass a broad range of actions, poses, relationships, and interactions as far as possible. At the same time, by adjusting the entities and masks, users can progressively refine and expand the event’s level of detail.
> > >
> > >
> > > ## Q2: Which visual attributes in the reference image are practically preserved and transferred
> > >
> > > **A2:** The transferred visual attributes:
> > > 1) shape details of reference entities
> > > 2) the structure and semantic layout
> > >
> > >
> > > The preserved visual attributes:
> > > 1) appearance details of reference entities and background
> > >
> > >
> > > Specifically, the appearance of each target entity is then determined by the entity switching path. Additionally, the entity masks can also further refine the customization quality through attention guidance and regulation, preventing appearance leakage between entities and ensuring a clearer generation of interactions and relationships among them.
> > >
> > >
> > >
> > > ## Q3: Specific attributes or aspects in which FreeEvent extends or outperforms existing studies.
> > >
> > > **A3:** Overall, FreeEvent is training-free, making it more efficient compared to existing customizing methods, which require training or fine-tuning. Additionally, it only requires a single reference image. Specifically, while FreeEvent does incorporate some existing methods, we have introduced several key improvements to better address the event customization task.
> > >
> > >
> > > 1) For event transferring, previous works [A, B] perform DDIM inversion on reference images to extract the self-attention maps and spatial features. In contrast, we directly use forward diffusion to the reference images for extracting and transferring. This further reduces the appearance leakage from the reference image and saves the inversion cost and additional model inference time. Specifically, on NVIDIA A100 GPU, this can save at least 2 minutes for each image. Besides, we have also compared the PnP [A] in our paper, as shown in Figure 4,11-15, PnP struggles to accurately generate the target entities, and suffer from severe appearance leakage from the reference image and between each target entity.
> > >
> > >
> > > 2) For entity switching, besides the cross-attention guidance, we further regulate the cross-attention map of each entity to avoid the appearance leakage between each target entity. The ablation results shown in Figure 5 and additional Table R2 has demonstrated the effectiveness of the cross-attention regulation process.
> > >
> > > As the first work in this direction, we hope our method can unveil new possibilities for more complex customization, meanwhile serving as a challenging baseline for future works.
> > >
> > >
> > >
> > > [A] Plug-and-play diffusion features for text-driven image-to-image translation. CVPR, 2023.
> > >
> > > [B] Masactrl: Tuning-free mutual self-attention control for consistent image synthesis and editing. ICCV, 2023.
> > >
> > >
> > >
> > > We sincerely appreciate your feedback. If you have any further questions or suggestions, please feel free to let us know.

---

### Decision · Program_Chairs · 2025-05-01

**Decision:**

Accept (poster)

**Comment:**

This paper proposes FreeEvent, a diffusion-based method for event-customized image synthesis. The core innovation is a disentanglement-recomposition framework enabling fine-grained event adaptation with minimal references. After the rebuttal, most of the concerns have been addressed, and reviewers raised their scores to accept or weakly accept. The work establishes a new paradigm for event-aware generation, warranting acceptance.